# Anti-SARS-CoV-2 Activity of Polysaccharides Extracted from *Halymenia floresii* and *Solieria chordalis* (Rhodophyta)

**DOI:** 10.3390/md21060348

**Published:** 2023-06-06

**Authors:** Clément Jousselin, Hugo Pliego-Cortés, Alexia Damour, Magali Garcia, Charles Bodet, Daniel Robledo, Nathalie Bourgougnon, Nicolas Lévêque

**Affiliations:** 1Laboratoire de Virologie et Mycobactériologie, Centre Hospitalier Universitaire, 86021 Poitiers, France; clement.jousselin@chu-poitiers.fr (C.J.); magali.garcia@chu-poitiers.fr (M.G.); 2Laboratoire Inflammation Tissus Epitheliaux et Cytokines, Université de Poitiers, 86073 Poitiers, France; alexia.damour@u-bordeaux.fr (A.D.); charles.bodet@univ-poitiers.fr (C.B.); 3Université Bretagne-Sud, EMR CNRS 6076, LBCM, IUEM, F-56000 Vannes, France; hugo-skyol.pliego-cortes@univ-ubs.fr (H.P.-C.); nathalie.bourgougnon@univ-ubs.fr (N.B.); 4Centro de Investigación y de Estudios Avanzados (CINVESTAV), Unidad Mérida, AP 73, Cordemex, Mérida 97310, Yucatán, Mexico; daniel.robledo@cinvestav.mx

**Keywords:** COVID-19, SARS-CoV-2, antiviral activity, red seaweed, sulfated polysaccharide

## Abstract

Even after hundreds of clinical trials, the search for new antivirals to treat COVID-19 is still relevant. Carrageenans are seaweed sulfated polysaccharides displaying antiviral activity against a wide range of respiratory viruses. The objective of this work was to study the antiviral properties of *Halymenia floresii* and *Solieria chordalis* carrageenans against SARS-CoV-2. Six polysaccharide fractions obtained from *H. floresii* and *S. chordalis* by Enzyme-Assisted Extraction (EAE) or Hot Water Extraction (HWE) were tested. The effect of carrageenan on viral replication was assessed during infection of human airway epithelial cells with a clinical strain of SARS-CoV-2. The addition of carrageenans at different times of the infection helped to determine their mechanism of antiviral action. The four polysaccharide fractions isolated from *H. floresii* displayed antiviral properties while the *S. chordalis* fractions did not. EAE-purified fractions caused a stronger reduction in viral RNA concentration. Their antiviral action is likely related to an inhibition of the virus attachment to the cell surface. This study confirms that carrageenans could be used as first-line treatment in the respiratory mucosa to inhibit the infection and transmission of SARS-CoV-2. Low production costs, low cytotoxicity, and a broad spectrum of antiviral properties constitute the main strengths of these natural molecules.

## 1. Introduction

Severe acute respiratory syndrome coronavirus (SARS-CoV)-2 emerged in Wuhan, China, in December 2019. It then spread all over the world constituting the second viral respiratory pandemic in the 21st century after the 2009/H1N1 influenza pandemic. At the time of writing, 756,411,740 confirmed cases worldwide and 6,842,462 deaths have been reported to the WHO (02/16/2023) [1]. SARS-CoV-2 is a member of the *Coronaviridae* family, the third to be involved in epidemic acute respiratory syndromes after SARS-CoV-1 and Middle East respiratory syndrome coronavirus (MERS-CoV), which emerged in 2002 and 2012, respectively [2,3]. This family is divided into three genera. SARS-CoV-2 belongs to the *Betacoronavirus* genus like SARS-CoV-1, MERS-CoV, and two species of seasonal coronaviruses (HcoV-OC43 and HcoV-HKU1) responsible for generally mild respiratory infections each winter. The two other coronaviruses infecting humans (HcoV-229E and HcoV-NL63) can be found in the *Alphacoronavirus* genus.

SARS-CoV-2, like other coronaviruses, is a single-stranded, positive-sense, enveloped RNA virus with a genome size of 39 Kilobases. It is mainly airborne, initially infecting the respiratory mucosa. Its replication cycle begins with the binding of the Spike protein envelope to the angiotensin-converting enzyme 2 (ACE2) receptor, strongly expressed at the surface of epithelial lung cells [4,5].

Infection with SARS-CoV-2, called COVID-19 (COronaVIrus Disease-19), leads to a wide range of clinical manifestations, from a simple cough with fever to a potentially life-threatening severe acute respiratory syndrome. Its pathophysiology is linked to the triggering of pro-inflammatory cytokine and chemokine production known as “cytokine storm” correlated with lung injuries, disease severity, and poor prognosis [6,7].

Hundreds of clinical trials have evaluated the efficacy of potential therapeutic strategies against SARS-CoV-2 involving new and repurposed antiviral drugs. An association of the two nirmatrelvir/ritonavir protease inhibitors, preventing the maturation of the viral polyprotein or remdesivir, a nucleotide analog prodrug developed against the Ebola virus, are among the few molecules that have proven their effectiveness [8,9]. Nevertheless, the rapid genetic evolution of SARS-CoV-2, through mutations and recombinations, is likely to cause the emergence of resistance to current antivirals. The search for new antiviral molecules to prevent and treat COVID-19 is therefore still relevant.

Seaweeds are an excellent source of bioactive compounds such as Carrageenans, which have been reported to have a variety of pharmacological activities [10]. Carrageenans are linear, sulfated galactans and one of the major constituents of red seaweed cell walls, representing 30% to 75% of the algal dry weight [11]. They are normally classified according to their structural characteristics, including sulfation-, pyruvate-, methyl-patterns, and the presence or absence of 3,6-anhydro bridges in α-linked galactose residues. They differ also from one another in degree of solubility and gelling properties. Carrageenans are therefore widely used for their gelling, thickening, and stabilizing properties in the food, cosmetic, and pharmaceutical industries, where they are used as pharmaceutical delivery vehicles to facilitate drug formulation or sustained drug release [12]. Kappa-, lambda-, and iota-carrageenans also display antiviral properties, which have been demonstrated against several enveloped viruses including a wide range of respiratory viruses such as paramyxoviruses, respiratory syncytial virus (RSV), influenza viruses (FluV), and seasonal coronaviruses [13,14,15,16,17,18,19,20,21]. Recently, Pliego-Cortés et al. (2022) demonstrated in vitro evidence of the antiviral activity of lambda- and iota-carrageenans extracted from stranding biomass of *Halymenia floresii* and *Solieria chordalis* against Herpes simplex virus 1 (HSV-1) [22].

*H. floresii* is a tropical red macroalga common to the coast of the Yucatan peninsula, Mexico. There, *H. floresii* dominates rocky substrates between 3 and 40 m where it grows up to 50 cm tall. While *H. floresii* stranding is common during the north wind season, this species has also been identified with high cultivation potential [23]. Its cultivation, under integrated multitrophic aquaculture systems, has shown the feasibility of producing biomass and, in a sustainable way, of extracting biomolecules such as carbohydrates, proteins, fatty acids, and mycosporine-like amino acids [24]. The native sulfated polysaccharides of *H. floresii* were previously studied by Freile-Pelegrin and colleagues. The authors reported that these polysaccharides seemed to be chemically and rheologically similar to the lambda-carrageenan family [25].

The Rhodophyta *S. chordalis* can form a veritable submarine meadow at 7–8 m depth. To develop, it needs to be in a turbid marine environment, semi-sheltered, and preferentially on a hard substrate. Since 2005, *S. chordalis* has been observed in the Gulf of Morbihan and in the Sarzeau peninsula where strandings have become more abundant between July and October. *S. chordalis* holds potential for biotechnological development since some of its polymers have been shown to possess immunological, hemagglutinic, and antiviral activities with practical application in biomedicine [26]. The genus *Solieria* contains iota-carrageenan as its main cell wall polysaccharide [27,28,29].

These two species have the advantage of being available in large volumes representing important sources for polysaccharide extraction. In this sense, the extraction efficacy must impact the potential utilization of the seaweed biomass. For instance, enzyme-assisted extraction (EAE) is a green environmentally friendly extraction method known for its high efficiency and for allowing the reduction or avoidance of the use of a solvent or alkali, which could induce chemical or structural modification (i.e., the cyclized derivatives in lambda-type carrageenans). Our research group showed the efficacy of the EAE procedure on native polysaccharide extraction yields from *Chondrus crispus*, *Solieria filiformis* and *chordalis*, and *H. floresii* using commercial enzymes, generally proteases or glucanases [22,30,31,32,33,34].

The objective of this work was to study the in vitro antiviral properties of lambda- and iota-carrageenan-rich polysaccharide fractions, extracted from *H. floresii* and *S. chordalis*, respectively, against the Wuhan Type of SARS-CoV-2. Evaluation of carrageenan treatment on viral replication was performed during kinetics of infection of a human lung carcinoma cell line (Calu-3 cells). *H. floresii* and *S. chordalis* fractions obtained using different extraction methods were tested. Finally, the addition of carrageenans at different times during the infection helped to determine their mechanism of antiviral action.

## 2. Results

### 2.1. Carrageenan Extraction from Halymenia floresii and Solieria chordalis

The polysaccharide fractions of *H. floresii* and *S. chordalis* were prepared as reported in a previous study [22]. Briefly, they were obtained by Enzyme-Assisted Extraction (EAE) or Hot Water Extraction (HWE), ethanol precipitation, and dialyzed (fractions 1.1 and 1.2), after which subsamples were purified by ion exchange resin (fractions 1.3 to 1.6) as shown in Table 1. All fractions were rich in neutral sugars (22.8–37% dw) and sulfate groups (9.4–18.6). The 3,6-Anhydrogalactose (3,6-AG) and protein content were lower in *H. floresii* (less than 2.7% dw of protein) than in *S. chordalis* (up to 8.8% dw of 3,6-AG). The full biochemical composition is presented in Appendix A. The main monosacharide in all fractions was galactose, representing up to 70% dw, with minor amounts of other monosaccharides (i.e., glucuronic acid, glucose, arabinose, etc.). The full monosaccharide composition of fractions 1.1 and 1.2 is presented in Appendix A, while that of 1.3 to 1.6 fractions has been reported in [22].

The molecular weight distribution of fractions 1.3 to 1.6 was determined by Gel Permeation Chromatography-Size Exclusion Chromatography (GPC-SEC) and is shown in Table 1. The weight-averaged molecular weight (Mw) ranged from 590 kDa (*S. chordalis*) to 1100 kDa (*H. floresii*) with a polydispersity index (PI) from 1.5 to 1.8. While the molecular weights of fractions 1.1 and 1.2 were reported as 1202 and 1569 kDa, respectively, according to Pliego-Cortés et al. [22] based on intrinsic viscosity.

The spectroscopic analysis of *H. floresii* and *S. chordalis* fractions made by the Fourier Transform Infrared (FTIR) showed spectra with typical bands for carrageenan identification, similar to those identified in commercial samples of lambda- or iota-carrageenans respectively used as references (Figure 1). All spectra presented the band of sulfate esters at 1210–1250 cm^−1^. The spectra of *H. floresii* were distinguished by the broadband at 830–820 cm^−1^, which corresponded to C-O-SO_3_ on C2 of galactose from G (G2S) and D units (D2S) concerning lambda-type carrageenans [22]. While *S. chordalis*’ spectra showed bands at 930 cm^−1^, belonging to C-O of 3,6-anhydrogalactose (3,6-AG) from DA unit, at 845 cm^−1^, corresponding to C-O-SO_3_ on C4 of galactose (G4S) from G unit, and at 805 cm^−1^, indicating the presence of C-O-SO3 on C2 (DA2S) of D unit, characteristic of iota-type carrageenans [26].

Since the polysaccharide fractions 1.1 and 1.3 rich in lambda-carrageenans obtained through EAE from *H. floresii* showed higher antiviral activity, their structural compositions were further explored by ^1^H Nuclear Magnetic Resonance (NMR). The spectra resulting from this analysis are presented in Figure 2. Both spectra showed chemical shifts (δ) corresponding to the anomeric hydrogens for the lambda-type carrageenan based on the published literature [22,35,36]. The δ = 5.50–5.52 ppm region was assigned as α-anomeric proton from D-unit (1,4-linked α-D-galatopyranose) corresponding to D2S,6S H-1. The signals at δ = 4.69 and 4.48 were identified as G-unit (1,3-linked β-galactopyranose) conforming to G2S H-1 and H-2, respectively. Additionally, two groups of chemical shifts may correspond to H-2/3/5 of D-unit (5.31–4.91 ppm) and H-3/5/6 of G-unit (3.53–3.83 ppm). The upfield signal around 1.4 ppm in the spectrum of fraction 1.3 may correspond to pyruvic acid acetal, as also reported in other carrageenans [28]. The high viscosity of the samples, even when hot, makes it difficult to obtain the ^13^C NMR spectra. ^1^H NMR and FTIR spectra of fractions 1.2 and 1.4 of *H. floresii* were reported on Pliego-Cortés et al. [22] in which the main polysaccharide was lambda-type carrageenans.

### 2.2. H. floresii and S. chordalis Fraction Cytotoxicity on Human Airway Epithelial Cells

The XTT cell viability assay showed a low percentage of toxicity (<10%) of *H. floresii* and *S. chordalis* fractions on Calu-3 cells, even at the highest concentration tested (40 µg/mL) (Figure 3). Moreover, neither alteration of normal cell morphology nor destruction of the cell monolayer was microscopically observed. The concentration of 10 µg/mL was retained for the following experiments.

### 2.3. Screening for Antiviral Activity of Carrageenans

Anti-SARS-CoV-2 in vitro activity of lambda- and iota-carrageenan-rich polysaccharide fractions extracted from *H. floresii* and *S. chordalis* was then assessed (Figure 4). The Calu-3 cells were infected with the virus at an MOI of 0.01 and treated with the fractions at a concentration of 10 µg/mL. Viral replication was assessed after 24 h of infection by measuring viral RNA by RT-qPCR in the cell monolayer (Figure 4A) and the culture supernatant (Figure 4B), and by titrating infectious viral particles in the supernatant (Figure 4C). As shown in Figure 4, all four *H. floresii* fractions displayed antiviral activity against SARS-CoV-2. However, fractions 1.1 and 1.3 (EAE) caused a stronger reduction in viral RNA concentration than fractions 1.2 and 1.4 (HWE), even though no significant differences were observed (Figure 4A,B). *H. floresii* fractions also showed greater antiviral activity than Heparin at the same concentration (10 µg/mL). Finally, *S. chordalis* fractions were not active on SARS-CoV-2 replication.

### 2.4. Time-of-Drug-Addition Study

The fractions 1.1 and 1.3 extracted from *H. floresii* by the protamex enzyme (EAE) and purified by dialysis or, by dialysis and then by ion exchange resin, respectively, showed the highest antiviral activity. They were retained to determine the mechanism of action of lambda-carrageenans against SARS-CoV-2 (Figure 5). These fractions were added at four different times of Calu-3 infection with SARS-CoV-2: pre-treatment of the cells with lambda-carrageenans 4 h before infection (−4 h), pre-treatment of the virus with carrageenans 1 h before infection (incubation −1 h), simultaneous addition of the lambda-carrageenans and the virus on the cells (simultaneous), and post-treatment of the cells with lambda-carrageenans 4 h after infection (+4 h). Viral replication was then assessed after 24 h of infection by RT-qPCR in cell monolayer and supernatant.

A reduction in viral replication was observed under all the conditions of the addition of the two fractions tested, whether in the cell monolayer or the cell culture supernatant. This reduction was nevertheless more marked, between 50 and 80%, and statistically significant when fractions 1.1 (B and C) and 1.3 (D and E) were added 4 h before infection, pre-incubated with the virus, or added 4 h post-infection. Simultaneous addition of the lambda-carrageenans and the virus to the cell culture, where contact duration between the virus and the cells with the polysaccharides was the shortest (1 h), was associated with lower inhibition of viral infection, by 40% on average. Finally, the activities of the two fractions 1.1 and 1.3 were compared with each other. Overall, no clear difference in activity between the two fractions could be observed. Moreover, the profile of viral replication inhibition according to the time of drug addition was similar between the two fractions suggesting an identical mechanism of action.

### 2.5. Impact of H. floresii 1.1 and 1.3 Fractions on SARS-CoV-2 Attachment to the Calu-3 Cells

After the time-of-drug-addition study, the ability of *H. floresii* polysaccharide-rich fractions in lambda-carrageenans to block the attachment of SARS-CoV-2 at the surface of target cells was investigated to explain their pre-fusion antiviral activity (Figure 6). Three protocols were tested: (i) incubation with the virus to saturate the viral envelope glycoproteins (SARS-CoV-2 +1.1 or +1.3); (ii) incubation of the fractions with the cell monolayer to saturate the cellular receptors (Cells +1.1 or +1.3); and (iii) a combination of the two protocols in which the cell monolayer, as well as the virus, were incubated separately with the fractions (SARS-CoV-2 + Cells +1.1 or +1.3) before coming into contact with each other. The viral adsorption assay showed a reduction of SARS-CoV-2 replication under the three protocols and for the two fractions tested. This inhibition of viral replication was, however, significant only when viral RNA was quantified in the cell culture supernatant (B and D).

## 3. Discussion

Each year, local winds and tides drive seaweeds to the shores, causing large strandings, which entail economic losses and the destruction of coastal marine habitats. On the Yucatan coast in Mexico, these strandings are observed between December and February, bringing high amounts of *Halymenia floresii*. In France, from August to October, *Solieria chordalis* biomass is stranded on the coastline of South Brittany. All these biomasses offer an opportunity to obtain sulfated polysaccharides such as carrageenans with many biological properties including antiviral activities.

In 2022, Pliego-Cortés et al. showed in vitro anti-herpetic activity of fractions of sulfated polysaccharides extracted from *Halymenia floresii* [22]. Semi-refined polysaccharide fractions, obtained by protamex enzyme (EAE) or hot water (HWE) extractions, ethanol precipitation and dialysis (6–8 kDa MWCO membrane) had, by cell viability, strong activity against HSV-1 with EC_50_ of 0.68 and 1.24 µg/mL, respectively, without toxicity on Vero cells. No previous study had reported this anti-herpetic activity of polysaccharide extracted from *H. floresii*, suggesting promising biomaterial, which could be used to further produce antiviral agents. Moreover, this study demonstrated the efficacy of the eco-friendly processes (Enzyme-Assisted Extraction) to increase the yield of extractions and prevent changes in the molecules, since polysaccharide extraction in red seaweeds is usually carried out with a strong hot-alkaline solution, which affects the structure and the size of the molecules, and also their biological properties [37]. Finally, *H. floresii* polysaccharides were found to be soluble in 0.3 M KCl solution and recovered from the solution using isopropanol, whereas *S. chordalis* extracts were insoluble and formed soft gels. These results, in agreement with previous studies, supported the predominant occurrence of carrageenans belonging to the lambda-family in *H. floresii* polysaccharide fractions while iota-carrageenans were the majority in *S. chordalis.* [22,38,39].

In direct line with this study, the objective of the present work was to evaluate the activity of lambda- and iota-carrageenan-rich polysaccharide fractions, respectively, extracted from *Halymenia floresii* and *Solieria chordalis* against SARS-CoV-2, a virus responsible for severe respiratory infections with a limited number of available treatments.

Our results showed that the four polysaccharide fractions isolated from *H. floresii* had antiviral properties against SARS-CoV-2 while *S. chordalis* fractions did not. Moreover, *H. floresii* fractions 1.1 and 1.3 (EAE) caused a stronger reduction in viral RNA concentration compared to fractions 1.2 and 1.4 (HWE). The structural characteristics of the carrageenans tested, linked not only to the producing seaweeds but also to their mode of extraction and purification, are likely to explain these results.

Carrageenans have a broad spectrum of antiviral activity [40]. However, their wide structural diversity and complexity make it challenging to establish a common structure-antiviral activity relationship [37]. Carrageenans have similar backbones based on D–galactose with strong negative charges, but they vary in their fine structure, which is related to antiviral activity. Previous studies have shown a positive correlation between antiviral activity and sulfate content. Yim et al. showed that crude polysaccharide from the red alga *Porphyra tenera,* with a non-detectable amount of sulfate ions, had no antiviral activity against SARS-CoV-2 pseudovirus [41]. Kang et al. reported that the polysaccharide extracted from the brown alga *Hizikia fusiforme* with the highest content of sulfate ions presented the highest anti-SARS-CoV-2 activity [42]. In our results, this correlation was observed only with fractions 1.1 and 1.2 of *H. floresii*. On the contrary, *S. chordalis* fractions with higher amounts of sulfate groups displayed lower antiviral activity. This could be explained insofar that, besides sulfate content, the distinct structural specificities, such as the sulfatation pattern in the backbone, the molecular weight, or chemical composition, are involved in the antiviral activities [37]. Based on this, the study of Kwon et al. showed that the complex polysaccharide RPI-27 obtained from *Saccharina japonica*, with higher molecular weight (100 kDa) than polysaccharide RPI-28 (12 kDa) and heparin (17 kDa), had the higher antiviral activity in inhibiting the binding of SARS-CoV-2 spike glycoprotein to the target cell [43,44]. Overall, the higher activity of RPI-fractions compared to heparin could be related to the multivalent interactions between the polysaccharide and the viral particle, since RPI-27 and RPI-28 are both highly branched whereas heparin is a linear polysaccharide. In our work, a higher degree of sulfation of galactose residues in agreement with a higher calculated molecular weight was assessed in fraction 1.1, which showed the highest antiviral activity. Similarly, *H. floresii* fractions were more efficient in inhibiting SARS-CoV-2 infection than unfractionated heparin at the same concentration (10 µg/mL) used as a positive control [45]. This could be explained by the smaller chain length and the lower molecular weight of heparin compared to lambda-carrageenans, which are the main constituents of the polysaccharide fractions extracted from *H. floresii*. Finally, the absence of anti-SARS-CoV-2 activity of fractions extracted from *S. chordalis*, rich in iota-carrageenans, assessed in the present work, contradicts previous data having demonstrated their efficacy in vitro as well as in nasal spray formulations [46,47,48]. It could be explained by the presence of high protein content in *S. chordalis* fractions linked to the Enzyme-Assisted Extraction method with protamex. This mixture of proteases is known to increase protein recovery, when extracting polysaccharide fractions from *S. chordalis*, with amounts ranging from 11.6 to 15.2% dry weight [30,31]. Proteins can be then involved in electrostatically-driven interactions with negatively-charged polysaccharides resulting in a coacervation and the formation of macromolecular complexes, significantly altering the structure of junction zones [49].

Time of addition experiments carried out subsequently did not reveal any significant differences between the conditions tested. Indeed, pre-treatment showed similar results on the inhibition of virus replication as the treatment after infection with SARS-CoV-2. These results suggest that carrageenans could be used both before the infection, prophylactically, as well as after the infection, therapeutically.

The antiviral action of the lambda-carrageenan-rich polysaccharide fractions extracted from *H. floresii* identified in our study seems to be related to the inhibition of the attachment of the virus to the host cell. This property of sulfated polysaccharides has been described. Jang et al. showed that lambda-carrageenans were effective against SARS-CoV-2 with EC_50_ 0.9 ± 1.1 µg/mL and suggested that the carrageenans target viral attachment to cell surface receptors and subsequently prevent viral entry [21]. Moreover, our results are in favor of the interaction of sulfated polysaccharides with viral envelope glycoproteins since virus incubation with lambda-carrageenans was more efficient in reducing virus binding than incubation of lambda-carrageenans with the cell monolayer. In 2020, the scientific team of Marinomed Biotech AG showed that iota-carrageenans can inhibit the cell entry of the SARS-CoV-2 spike pseudo-typed lentivirus in a dose-dependent manner. SARS-CoV-2 spike pseudo-typed lentivirus particles were efficiently neutralized with an IC_50_ value of 2.6 μg/mL iota-carrageenans [46]. This pre-fusion activity, before the entry of the virus into the cell, is likely linked to the high molecular weight of carrageenans which hardly penetrate into the cell and are more likely to line its surface.

Due to their size and their mode of action, the nasal spray formulation of carrageenans is entirely consistent with the objective of protecting the respiratory mucosa by preventing the attachment of the virus. Several nasal sprays containing carrageenans have been tested against respiratory viruses. According to Koenighofer et al., the administration of a carrageenan nasal spray, in children as well as in adults suffering from the respiratory virus-confirmed common cold, reduced the duration of disease, increased viral clearance, and reduced relapses of symptoms [50]. Jang et al. showed that the use of commercial lambda-carrageenans as nasal drops in mice reduced weight loss resulting from influenza viral infection and prevented infection-related death [21]. Regarding SARS-CoV-2, Schütz et al. showed that both nasal and oral sprays containing kappa-carrageenans inhibited its replication in human airway epithelial cells [20]. Moreover, the Xylitol^®^ nasal spray containing iota-carrageenans has been found to prevent SARS-CoV-2 infection in vitro, with an IC_50_ < 6.0 µg/mL [47]. In vivo, Figueroa and colleagues demonstrated that iota-Carrageenan nasal spray is an effective prophylaxis to prevent SARS-CoV-2 infection in healthcare workers managing patients with COVID-19 disease [48].

In conclusion, this study confirmed that carrageenans have the potential to be the first line of defense to inhibit the infection and transmission of SARS-CoV-2. They can block the initial stages of infection consisting of virus attachment to the target cell. Low production costs, low cytotoxicity, and a broad spectrum of antiviral properties constitute the main strengths of these natural molecules.

## 4. Materials and Methods

### 4.1. Algal Material and Extraction of Sulfated Polysaccharides

The red seaweeds *Halymenia floresii* (Rhodophyta, Halymeniales) and *Solieria chordalis* (Rhodophyta, Gigartinales) were collected from algal stranding events. The former were collected on the coast of Sisal in Yucatan, Mexico, and the latter on the littoral zone of Saint Gildas de Rhuys in Brittany, France. The fresh biomass was cleaned, freeze-dried, and ground into a fine powder (raw biomass).

Polysaccharide extraction was carried out according to Pliego-Cortés et al. [22]. Briefly, a sample of depigmented, defatted, and dried raw biomass (20 g) was extracted in distilled water, following an Enzyme-Assisted Extraction (EAE) with 5.0% (*w*/*w*) of Protamex^®^ enzyme (Novozymes, Bagsværd, Denmark), for 3 h at 50 °C with constant stirring, followed by 15 min at 90 °C to denature the enzyme. On a parallel track, hot-Water Extraction (HWE) was carried out as mentioned above but without enzymatic treatment. All extractions were performed in triplicate. The extracts were collected by centrifugation (5000× *g* for 10 min), lyophilized, and re-dissolved in distilled water and the polysaccharides were precipitated with four volumes of 99% cold EtOH (*v*/*v*) at 4 °C overnight. Precipitates were collected by centrifugation (7000× *g* for 20 min), washed out with absolute EtOH and acetone, air-dried, and milled into fine powder. The powder was re-suspended in distilled water (20 mg/mL) and dialyzed against distilled water for 3 days at 4 °C. Polysaccharides were collected with EtOH as described above. These samples were named Fraction 1.1 (from EAE) and Fraction 1.2 (from HWE) for *H. floresii*. The samples were mainly composed of lambda-carrageenan with 24.7% dry weight (dw) of neutral sugars, 17.2% dw of sulfate groups, 1.9% of uronics acids, 2.7% of protein, and 1.0% of 3,6-Anhydrogalactose. The full details on the composition and structure are found in Pliego-Cortés et al. [22]. While the sulfated polysaccharides of *S. chordalis* EAE and HWE were composed of iota-carrageenan with 27.2 and 26.1% dw of neutral sugars, 12.9 and 10.2% dw of sulfate groups, 5.4 and 6.2% of uronics acids, 12.7 and 10.2% of protein and 9.0 and 10.1% of 3,6-Anhydrogalactose, respectively.

A sample of each species was further purified by ion exchange chromatography. The sample was dissolved in mQ water, filtered at 1 µm and loaded onto an XK 26 Pharmacia column (50 × 600 mm) packed with DEAE-Sepharose fast-flow anionic resin (GE Healthcare, Uppsala, Sweden), which was eluted at 4 mL/min with increasing concentrations of NaCl solution until 100% 1 M. Full details of the process can be found in Pliego-Cortés et al. [22]. In this study, Fraction 2 (F2) was used for the antiviral assays. These purified samples were denoted as Fraction 1.3 (from EAE) and Fraction 1.4 (from HWE) for *H. floresii*. Whereas, for *S. chordalis* samples were named fraction 1.5 (from EAE) and fraction 1.6 (from HWE), respectively. All purifications were done in triplicate. The full biochemical composition, including the monosaccharide, of these fractions is reported in Pliego-Cortés et al. [22].

### 4.2. Chemical and Structural Composition of Purified Polysaccharides

The molecular weight distributions for samples 1.3 to 1.6 were determined by Gel Permeation Chromatography/Steric Exclusion Chromatography system (Agilent Infinity II, Shropshire, UK) attached to a UHPLC Ultimate 3000 (Thermo, Germering, Germany) using a guard column PWXL and column TSK gel G6000PWXL (30 cm × 7.8 mm, 13 µm). Elution was stable with a 0.1 M sodium nitrate solution at 0.7 mL/min. Polysaccharides were detected by the Refractive Index. Chromatograms were analyzed by Chromeleon software v. 7.2 (Thermo, MA, USA) based on the standard curve of dextran with molecular weights ranging from 1000 to 670,000 Da (Sigma-Aldrich, Saint louis, MO, USA). The number-averaged molecular weight (Mn), weight-averaged molecular weight (Mw), and polydispersity index (I) were calculated by Agilent sofware v. 2.2 (Agilent, Waldbronn, Germany). The molecular weight of samples 1.1 (1202 kDa) and 1.2 (1569 kDa) were reported by Pliego-Cortés et al. [22], and were determined by the intrinsic viscosity of the polysaccharides. The Fourier Transform Infrared (FTIR) spectroscopic analysis was performed using a Nicolet iS5 spectrometer (Thermo, Madison, WI, USA), equipped with a universal attenuated total reflectance (ATR) sampling device containing a diamond crystal. Samples obtained by EAE (1.1, 1.3, 1.5 and 1.6) of both species, as well as the commercial lambda-carrageenans Satiagum BDC 20, and iota-carrageenans Satiagel DF 52 (SKW Biosystems, Boulogne, France) included as references, were recorded in transmission mode at room temperature from 500 to 4000 cm^−1^, with 16 scans and a resolution of 4 cm^−1^. Background spectra of air were scanned before analyzing samples. The FTIR spectra were acquired and processed by Omnic 9.3.32 software (Thermo Scientific, Illkirch, France). ^1^H Nuclear magnetic resonance (^1^H NMR) spectra for fraction 1.1 and 1.3 from EAE of *H. floresii* were recorded using a Bruker Avance III HD 500 spectrometer (Bruker, Wissembourg, France) at 343 or 353 °K. The instrument was equipped with an indirect 5 mm probe head BBO 1H/{BB}. Samples were solubilized in 700 µL of D_2_O 99.96% and spectra were performed according to Bruker’s pulse program recommendations with standard pulse sequence, delay of 2 s, and a 30° pulse. Chemical shifts (δ) were expressed in ppm relative to tetramethylsilane (TMS) as an external reference.

### 4.3. Cell Lines

The Calu-3 cells (human lung carcinoma cell line) were maintained in complete Dulbecco’s modified Eagle medium/F12 Nutrient Mixture (DMEM/F12 1:1 +GlutaMax) (Gibco, Waltham, MA, USA) supplemented with 5% fetal bovine serum (FBS). Simian Vero E6 cells were maintained in DMEM (Gibco) supplemented with 5% FBS. Both cells were cultivated at 37 °C in a humidified atmosphere containing 5% CO_2_.

### 4.4. SARS-CoV-2 Strain and Production

The Wuhan Type SARS-CoV-2 virus was isolated at Poitiers university hospital from nasopharyngeal sample of a patient infected in March 2020. SARS-CoV-2 virus was isolated and amplified in Vero E6 cells in DMEM supplemented with 2% FBS, 100 U/mL Penicillin, 100 µg/mL streptomycin, and 2 µg/mL Amphotericin B. The supernatant was collected at 3 days post-infection, cleared by centrifugation, and then diluted in conservation medium (0.5 M Sucrose and 50 mM Hepes). Aliquots were frozen at −80 °C until use. Their viral titer assessed by median Tissue Culture Infectious Dose (TCID_50_) in Vero E6 cells was 10^5.98^ TCID_50_/mL.

### 4.5. Assessment of In Vitro Halymenia floresii and Solieria chordalis Fractions Cytotoxicity

Calu-3 cells were cultured in 96-well plates at 6 × 10^4^ cells per well in 0.1 mL of DMEM/F12 up to 80% confluence before being incubated with *H. floresii* and *S. chordalis* fractions at concentrations of 5, 10, 20, and 40 µg/mL. Cell viability was assessed after 24 h incubation using the cell proliferation kit II (XTT; Roche Diagnostics, Meylan, France) in accordance with the manufacturer’s instructions and compared with non-treated cells. All experiments were performed in duplicate.

### 4.6. Evaluation of the Antiviral Properties of Halymenia floresii and Solieria chordalis Fractions

The antiviral properties of *H. floresii* and *S. chordalis* fractions were first assessed during a 24 h SARS-CoV-2 infection of human airway epithelial cells. Calu-3 were cultured in 24-well plates at 4 × 10^5^ cells per well in 1mL of DMEM/F12. At 80% confluence, they were treated with 10 µg/mL of *H. floresii* or *S. chordalis* polysaccharide fractions or Heparin sodium salt from porcine intestinal mucosa (Sigma-Aldrich) 1 h and infected with SARS-CoV-2 at a MOI of 0.01. After 1 h incubation with the virus and the fractions or Heparin, the cell monolayer was rinsed twice with 500 µL of DMEM/F12. Finally, 1 mL of fresh medium containing the initial concentrations of *H. floresii*, *S. chordalis* or Heparin was added for additional incubation of 23 h. At 24 h post-infection, cell monolayers and cell culture supernatants were harvested for viral RNA extraction and viral load quantification by RT-qPCR. Virus titer was assessed in the supernatant by end point dilution on Vero cells.

### 4.7. Time-of-Drug-Addition Studies of Halymenia floresii Fractions

To determine whether *H. floresii* fractions 1.1 and 1.3 act on entry or post-entry steps of the SARS-CoV-2 replication cycle, four times of addition of the fractions during Calu-3 cells 24 h infection kinetics were tested: (i) 4 h before a 1 h-infection and a 23 h-incubation with fresh medium without fractions; (ii) pre-incubation of the virus and the fractions 1 h at 37 °C before addition of the mixture to the cell monolayer for 1 h of infection before removal of the supernatant and 23 h incubation with fresh medium without fractions; (iii) simultaneous addition of the virus and the fractions to the cells for 1 h before removal of the supernatant and 23 h incubation with fresh medium without fractions and (iv) 4 h post-infection then maintained until the end of the 24 h infection. Calu-3 were cultured in 24-well plates at 4 × 10^5^ cells per well. At 80% confluence, cells were infected at an MOI of 0.01 and fractions 1.1 and 1.3 were used at a single concentration of 10 µg/mL. At the end of the 24 h incubation time, cell monolayers and supernatants were collected for viral quantification via RT-qPCR.

### 4.8. Viral Load Quantification by RT-qPCR

For intracellular viral RNA quantification and evaluation of the host inflammatory response, total RNA was extracted from the Calu-3 monolayer using the NucleoSpin RNA extraction kit (Macherey-Nagel, Illkirch, France) in accordance with the manufacturer’s instructions. For viral RNA quantification in Calu-3 supernatant, 140 μL of cell culture medium were extracted using QIAmp viral RNA kit (Qiagen, Courtaboeuf, France) according to manufacturer’s recommendations and eluted in 50 µL of H_2_O. RNA concentration in cell monolayer extract was determined using the Nanodrop 2000 spectrophotometer (ThermoFisher Scientific, Waltham, MA, USA). One nanogram of total RNA extracted from cell monolayer and 10 µL of RNA extracted from cell supernatant were analyzed. The viral RNA was quantified by TaqPath™ COVID-19 CE-IVD RT-PCR Kit (ThermoFisher Scientific). Briefly, RT-qPCR reactions were performed in duplicate, in 15 µL of RT-PCR reaction mix and 10µL of RNA on QuantStudioᵀᴹ System (ThermoFisher Scientific). After Reverse Transcription at 55 °C for 20 min and initial denaturation at 95 °C for 2 min, PCR conditions were 15 s at 95 °C, 45 s at 55 °C and 15 s at 72 °C for 45 repeats.

The calibration range was determined using a viral RNA purified from a titrated viral suspension. Viral RNA was diluted to obtain a calibration range allowing for quantification of viral load from 10 to 10^4^ TCID_50_/10 µL.

### 4.9. Virus Titration by End-Point Dilution Assay

Vero E6 cells were seeded in 96-well plates the day before titration at the rate of 2500 cells/well in DMEM supplemented with 2% FBS. The viral suspension was successively diluted from 10^−1^ to 10^−9^ in DMEM medium supplemented with 2% FBS. Then, 100 µL of each dilution were deposited in 6 wells of a row. A reading was performed after 96 h of incubation at 37 °C in an atmosphere containing 5% CO_2_. The wells in which the cells had a cytopathic effect were considered positive for viral infection. The titer of the viral suspension was then determined using Kärber’s method for assessing the TCID_50_/mL.

### 4.10. Viral Adsorption Assay

Calu-3 cells were cultured in 24-well plates at 4 × 10^5^ cells per well in 1 mL DMEM/F12 (Gibco) supplemented with 5% FBS at 37 °C in a humidified atmosphere under 5% CO_2_. At 80% confluence, three protocols of saturation with *H. floresii* 1.1 and 1.3 fractions at a concentration of 10 μg/mL were tested: (i) incubation with the virus to saturate the viral envelope glycoproteins (SARS-CoV-2 +1.1 or +1.3); (ii) incubation of the fractions with the cell monolayer to saturate the cellular receptors (Cells +1.1 or +1.3); and (iii) a combination of the two protocols in which the cell monolayer as well as the virus were incubated separately with the fractions (SARS-CoV-2 + Cells +1.1 or +1.3). Virus at MOI of 0.01 and cells were then put into contact to allow viral adsorption. To lock the membrane structure and prevent viral internalization by endocytosis, these first two steps were carried out 1 h at 4 °C. The cells were then washed twice and incubated for 24 h at 37 °C before measuring the viral load by RT-qPCR in the monolayer and the supernatant.

### 4.11. Transcriptomic Analysis of the Human Airway Epithelial Calu-3 Cells Innate Immune Response

First, RNA of infected Calu-3 was reverse-transcribed using SuperScript II kit (Invitrogen, Life Technologies, Toulouse, France) according to manufacturer’s instructions. Secondly, quantitative real-time PCR was performed using 1 μM forward and reverse primers, designed using Primer 3 software (bioinfo.ut.ee/primer3-0.4.0/), and 12.5 ng of cDNA template in a total volume of 10 µL. PCR conditions were as follows: 5 min at 95 °C, 40 amplification cycles comprising 20 s at 95 °C, 15 s at 64 °C and 20 s at 72 °C. Samples were normalized with regard to two independent control housekeeping genes (ribosomal protein L13a and 28S rRNA gene) and reported according to the ΔΔCT method as RNA fold increase: 2^ΔΔCT^ = 2^ΔCT sample − ΔCT reference^.

### 4.12. Statistical Analysis

Results were analyzed by GraphPad Prism, version 5 (GraphPad Software, La Jolla, CA, USA). The statistical significance of the difference between two groups was evaluated by one-way non-parametric ANOVA Kruskal-Wallis. Differences were considered to be significant at *p* < 0.05.

## Figures and Tables

**Figure 1 marinedrugs-21-00348-f001:**
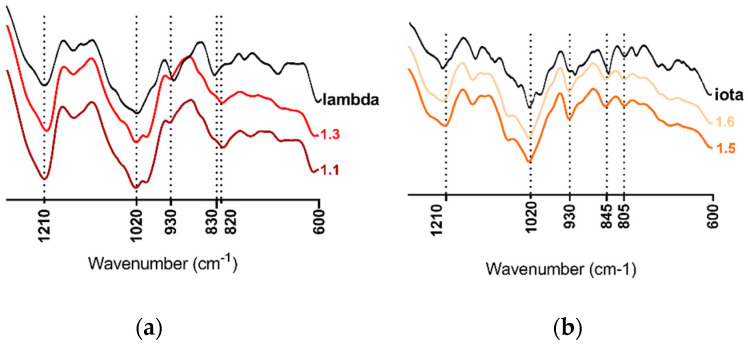
FTIR spectra of the polysaccharide fractions 1.1 and 1.3 of *Halymenia floresii* compared to commercial lambda-carrageenans (**a**), and fractions 1.5 and 1.6 of *Solieria chordalis* compared to commercial iota-carrageenans (**b**).

**Figure 2 marinedrugs-21-00348-f002:**
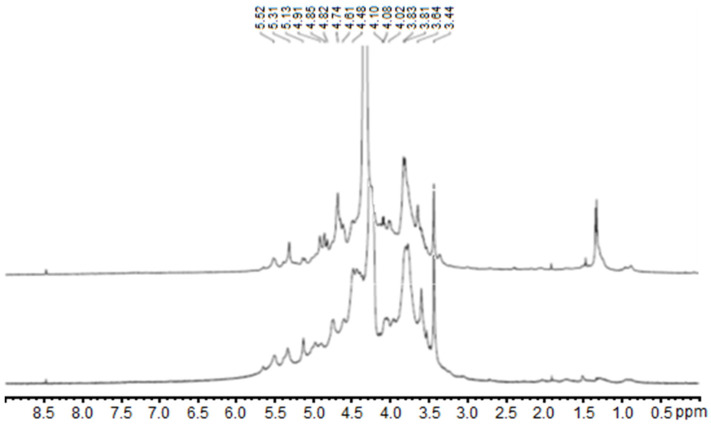
^1^H NMR spectra of the polysaccharide fractions 1.1 (bottom) and 1.3 (top) from Halymenia floresii, corresponding to the dialyzed and ion exchange purified fractions, respectively.

**Figure 3 marinedrugs-21-00348-f003:**
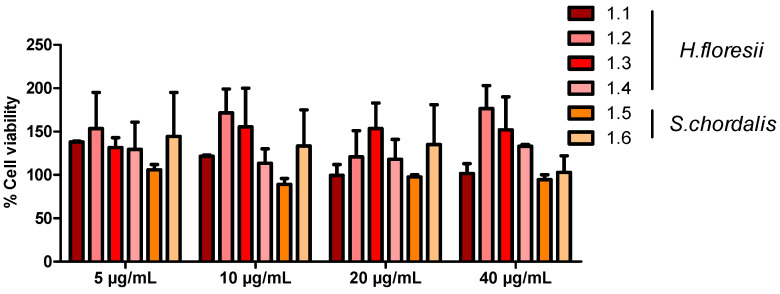
Determination of toxicity of Halymenia floresii and Solieria chordalis fractions on human airway epithelial cells. Percentage of viability of Calu-3 cells after 24 h of incubation with 4 fractions of *H. floresii* (1.1 to 1.4) and 2 fractions of *S. chordalis* (1.5 and 1.6) at concentrations ranging from 5 to 40 μg/mL was assessed with the XTT assay and compared to untreated cells. Mean ± SEM of two independent experiments, each in duplicate, are represented here.

**Figure 4 marinedrugs-21-00348-f004:**
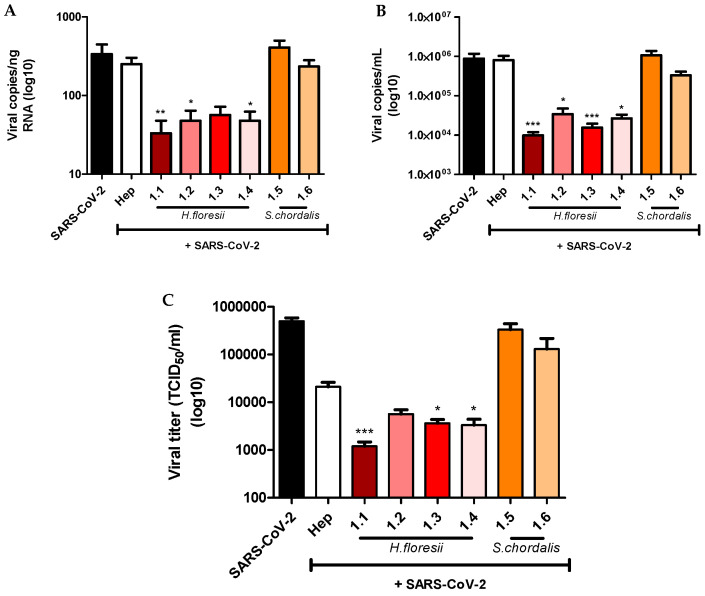
Effect of *H. floresii* and *S. chordalis* fractions on SARS-CoV-2 replication. Calu-3 cells were incubated 1 h with *H. floresii* and *S. chordalis* fractions or heparin, at a single concentration of 10 µg/mL and then infected with SARS-CoV-2 at 0.01 MOI. After 1 h of incubation with the polysaccharide and the virus, cell monolayer was washed and fresh medium containing the polysaccharide alone was added. Viral load was measured at 24 h post-infection in the cell monolayer (**A**) and the supernatant (**B**) while virus titer was assessed in the supernatant by end-point dilution assay on Vero cells (**C**). Mean ± SEM of five independent experiments in duplicate are represented here. Statistical significance was obtained compared to SARS-CoV-2. * *p* < 0.05, ** *p* < 0.01 and *** *p* < 0.001. Hep: Heparin.

**Figure 5 marinedrugs-21-00348-f005:**
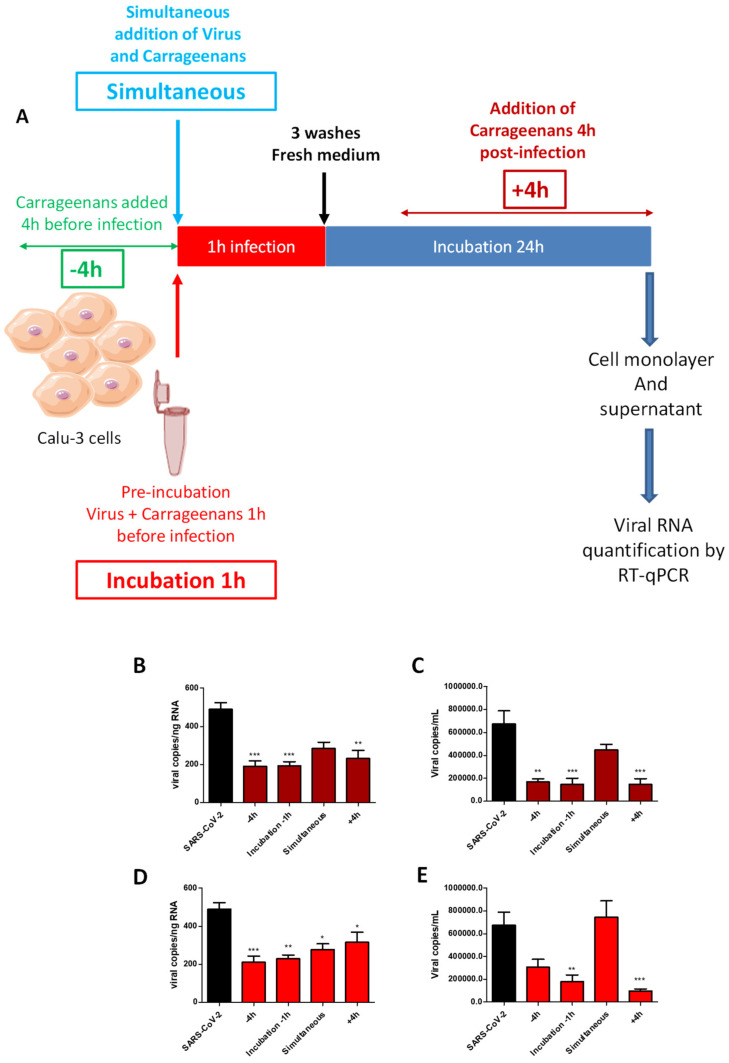
Time-of-drug-addition study of *H. floresii* fractions. Four conditions were tested to determine the viral replication step inhibited by the fractions 1.1 and 1.3 of lambda-carrageenans extracted from *H. floresii*. Cells were (i) incubated 4 h before infection with the polysaccharides (−4 h), (ii) infected with a virus + polysaccharide mixture pre-incubated for 1 h at 37 °C (incubation −1 h), (iii) infected with the virus added simultaneously to the polysaccharide (simultaneous) and (iv) infected and treated with the polysaccharide 4 h post-infection (+4 h) as schematically depicted in (**A**). Viral replication was assessed at 24 h post-infection in the cell monolayers (**B**,**D**) as well as in cell culture supernatants (**C**,**E**) treated with fractions 1.1 (**B**,**C**) and 1.3 (**D**,**E**). Mean ± SEM of five independent experiments, each in duplicate, are represented here. Statistical significance was obtained compared to SARS-CoV-2. * *p* < 0.05, ** *p* < 0.01 and *** *p* < 0.001.

**Figure 6 marinedrugs-21-00348-f006:**
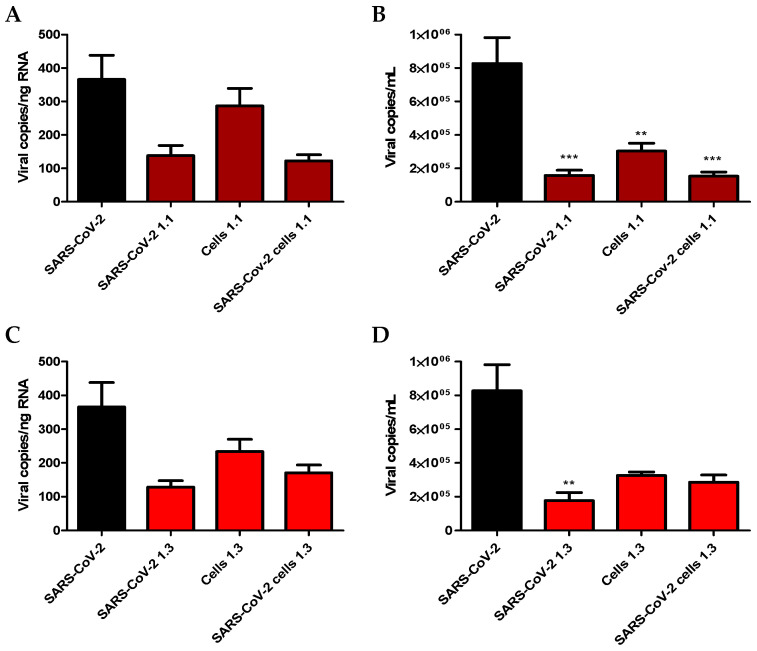
Effect of *H. floresii* lambda-carrageenans on pre-fusion steps of SARS-CoV-2 infection of human lung carcinoma cell line Calu-3. The cell monolayer, the virus (MOI of 0.01), or both, were independently incubated for 1 h at 4 °C with 10 μg/mL of carrageenan fractions 1.1 (**A**,**B**) and 1.3 (**C**,**D**). Virus and cells were then placed into contact for an additional 1 h at 4 °C. Cells were washed and fresh medium was added for 24 h incubation at 37 °C. Viral load was measured by RT-qPCR at 24 h post-infection in the cell monolayer (**A**,**C**) and the supernatant (**B**,**D**) reflecting attachment of the virus to the cell surface at the initial step of the experiment at 4 °C. The data presented here are the mean ± SEM of five independent experiments in duplicate. Statistical significance was obtained compared to SARS-CoV-2. ** *p* < 0.01 and *** *p* < 0.001.

**Table 1 marinedrugs-21-00348-t001:** Extraction, purification, and molecular weight characterization for the different fractions obtained from the red seaweeds *Halymenia floresii* and *Solieria chordalis*.

Fraction	Seaweed Species	Extraction	Purification	Mn ^1^ (kDa)	Mw ^1^ (kDa)	PI ^1^	AMw ^2^ (kDa)
1.1	*Halymenia* *floresii*	EAE	Dialysis	/	/		1202
1.2		HWE	Dialysis	/	/		1569
1.3		EAE	Ion exchange	500.1	911.8	1.8	
1.4		HWE	Ion exchange	624.7	1106.1	1.7	
1.5	*Solieria chordalis*	EAE	Ion exchange	384.1	596.3	1.5	
1.6		HWE	Ion exchange	363.3	622.6	1.7	

^1^ Determined by the refractive index in a GPC/SEC system. ^2^ Calculated by intrinsic viscosity. EAE: Enzyme-Assisted Extraction with protamex; HWE: Hot Water Extraction; Mn: number-averaged molecular weight; Mw: weight-averaged molecular weight; PI: polydispersity index; /: non-determined (peaks outside of the standard curve). AMw: weight-averaged molecular mass.

## Data Availability

The original data presented in the study are included in the article/Appendix A; further inquiries can be directed to the corresponding author.

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
