# Peer review of "Anti-SARS-CoV-2 Activity of Polysaccharides Extracted from Halymenia floresii and Solieria chordalis (Rhodophyta)"

_marinedrugs, 2023, doi:10.3390/md21060348_

Round 1
Reviewer 1 Report
In this manuscript the authors intent to investigate the antiviral activity of carrageenans extracted from Halymenia floresii and Solieria chordalis against SARS-CoV-2. With the arising of the SARS-CoV-2 pandemic growing evidence manifested in the field that carrageenans, particularly iota-carrageenan, exhibit antiviral activity, demonstrated by different laboratories both, in vitro and in vivo. Furthermore, the antiviral activity of iota-carrageenan has been demonstrated before for a variety of respiratory viruses, including SARS-CoV-2. Also for lambda-carrageenan, an antiviral activity against SARS-CoV-2 was reported although proven less effective than iota-carrageenan. While iota-carrageenan cannot enter cells because of its molecular size, other like lambda carrageenan do penetrate cells and exert numbers of biological activities, among them immune activation. Most importantly, a multicenter, randomized, double-blinded, placebo-controlled clinical study revealed that an iota-carrageenan containing nasal spray exhibits prophylactic efficacy in preventing SARS-CoV-2 infection in healthcare workers caring for patients with COVID-19 disease. Another clinical trial investigated a nasal spray containing both, Ivermectin and iota-carrageenan showed reduction in COVID-19 as well as of disease severity.
While this manuscript might add additional information to the growing knowledge about antiviral activities of polysaccharides, the almost complete disrespect towards state of the published work prohibits any consideration about publishing this work as it was submitted for peer review and needs major revisions. Finally, the manuscript should be shortened to a note, as the only new information considers the anti-SARS-CoV-2 activity from the red algae Halymenia floresii and Solieria chordalis, which was not shown before, while nothing is new about the activities of iota- and lambda-carrageenans reported in this submission. Moreover, the quality as well as the interpretation of the data requires significant improvement.
Specific comments
Major
1. As depicted in figure 1 and 2 the authors conducted spectroscopic and 1H NMR analysis. However, there are some issues with these analyses. First, the extracted fractions from Halymenia floresii and Solieria chordalis represent not an homogenous composition but rather a mixture of anything, among evidently iota- and lambda-carrageenan. Second, FTIR and 1H NMR analysis are not suitable to absolutely define the molecular structure of polymers like carrageenans, particularly if heterogeneous fractions are investigated. In fact, the authors could just make valuable conclusions about the antiviral efficacy of carrageenans, when there is a clear molecular characterization of the used fractions. At the moment they cannot define the nature and the content of the polymers in the extracts investigated.
2. Throughout the whole manuscript, the authors mainly use the term carrageenan, when they describe their own and also published work about the antiviral activity of carrageenans. E.g. they stated on page 2, line 70-75: “They also display antiviral properties, which have been demonstrated against several enveloped viruses including a wide range of respiratory viruses such as paramyxoviruses, respiratory syncytial virus (RSV), influenza viruses (FluV) and seasonal coronaviruses [13-21]. Recently, Pliego-Cortés et al. (2022) added in vitro evidence of the antiviral activity of carrageenans extracted from stranding biomass of Halymenia floresii and Solieria chordalis against Herpes simplex virus 1 (HSV-1) [22].” The authors should clearly indicate throughout the manuscript which type of carrageenan was used in these published studies and also what kind of carrageenans are used in the their own work. In this regard, it is well known from previous published work that there are significant differences in the antiviral efficacy of the different carrageenan-types. In general, iota-carrageenan shows a better antiviral effect than lambda- or kappa-carrageenan.
In this context, it is also surprising that the extract from S.cordalis, which, according to the authors, contains iota-carrageenan does not have any antiviral effect, whereas H. floresii, which contains lambda-carrageenan inhibits virus replication. This is in contrast to the previously published work, especially regarding SARS-CoV-2. This issue is also not discussed by the authors correctly. They stated in the discussion, page 9, line 303-304: “H. floresii contain mainly lambda-carrageenans and S. chordalis iota-carrageenans, the former being more active.” This statement is in contrast to their own work, were they show that iota-carrageenan is less active than lambda-carrageenan (Fig. 4).
3. In figure 3 the authors analyzed the influence of the used fractions from red algae on the cell viability. The results of these experiments are somehow confusing. First, it seems that the treatment of the Calu-3 cells with the different fractions has a clear influence on the cell viability, inasmuch as the viability is increasing up to 200%. Have the authors any explanation for this observation? Moreover, the presented results just represents the results of two independent experiments. It is common practice in science to perform at least three independent experiments, also in order to be able to calculate significance. Second, the author should use a positive control for the induction of apoptosis, e.g. staurosporine.
4. In figure 4 the authors used Heparin as a control, which is inappropriate. In their work, they mainly intend to show, that lambda- or iota-carrageenan from Halymenia floresii and Solieria chordalis inhibits the replication of SARS-CoV-2. It is published in several papers from several groups, that both, iota- and lambda-carrageenan exhibits antiviral activity against SARS-CoV-2. Thus, the well-characterized and – in addition - worldwide available carrageenans iota and lambda should be used as appropriate controls to demonstrate the efficacy in the inhibition of SARS-CoV-2 replication.
5. In figure 4 the authors measured viral RNA in the cell supernatant as well as in the cell monolayer. Why did they measured the content of the viral RNA intracellularly? Moreover, the labeling of the figures is confusing. In Figure 4 A, the authors used the term “viral copies/ng RNA” in figure 4 B, they used “viral copies/mL” and in figure 4 C: “Viral titer”. What do the authors mean with the term “viral titer”? Overall, the authors should choose a comprehensible and unique labeling.
6. In figure 5 the authors conducted time of addition experiments. First of all, it would be advantageous for the fast understanding of this series of experiments to integrate a schematic depiction of the different experimental setups in the figure.
Coming to the results of this figure, there were no big differences between the different treatment options. Clear conclusions from the authors according to these results are missing, because they might be impossible. The authors do not write anything in the abstract nor in the results section about their conclusions regarding the results of these experiments. Since the pre-treatment shows a very similar result on the inhibition of virus replication as the treatment after infection with SARS-CoV-2, it might be concluded that carrageenans could be used in both cases: before an infection, i. e. prophylactically, and after an infection, i. e. therapeutically. However, this has not been shown before, even not in the clinical trials.
Even worse, in the paragraph in the results section concerning these experiments, the authors wrote on page 6, line 205-207: “Finally, the activities of the two fractions 1.1 and 1.3 were compared with each other. Overall, fraction 1.1 exhibited significantly higher antiviral activity against SARS-CoV-2 than fraction 1.3.”. However, the authors did not directly compare the different fractions, at least not within the depicted figure 5, thus they could not conclude from this figure, that fraction 1.1 indeed exhibits significant higher antiviral activity compared with fraction 1.3.
7. In figure 6 the authors intend to analyze the immunomodulatory properties of the H. floresii fractions. As stated by the authors, these fractions contain lambda-carrageenan. It was published previously in countless publications that lambda carrageenan has immunomodulatory properties, e.g. it could induce inflammation, which is in a clear contrast to the results of these manuscript. Thus, the authors should conduct more thorough tests or entirely remove this figures and the conclusions drawn from these experiments.
8. In figure 7 the authors wanted to investigate if the different fractions of H.floresii have any influence on the SARS-CoV-2 attachment to the cells. Thereby, they concluded from their experiments on page 8, line 243-244: ”The viral adsorption assay showed a reduction of SARS-CoV-2 replication under the three protocols and for the two fractions tested.” Looking at the figure, it seem that in the most cases there is no significant reduction of virus replication. Thus, the authors could only make pure speculations, that the binding of the virus to the cells is affected. This conclusion is not supported by the results of the conducted experiments. The authors should remove these speculations or perform experiments, which clearly support their conclusions.
Minor
1. Within the title the authors state: “Anti-SARS-CoV-2 activity of polysaccharides extracted from Halymenia floresii and Solieria chordalis (Rhodophyta).” This title is misleading as there are different variants of concern of SARS-CoV-2 (Alpha, Beta, Gamma, Delta and Omicron). However, the authors only investigated the influence of different polysaccharides on the replication of SARS-CoV-2 Wuhan Type, this should be clearly indicated in the title.
2. In the introduction the authors should be clearer in the scientific classification of SARS-CoV-2 as well as the other endemic coronaviruses. Some belongs to the beta-coronaviruses and some of them to the alpha-coronaviruses.
3. In the introduction page 2, line 56-59 the authors stated: “Nevertheless, the rapid genetic evolution of SARS-CoV-2, through mutations and recombinations, is likely to cause the emergence of resistance to current antivirals. The search for new antiviral molecules to prevent and treat COVID-19 is therefore still relevant.” This is a wrong argument, as, except for monoclonal antibodies, no resistance mutations have been reported so far for any antiviral against SARS-CoV-2.
4. On page 2, line 92-94 the authors stated: “Evaluation of carrageenan treatment on viral replication was performed during kinetics of infection of human airway epithelial Calu-3 cells.” This sounds like they used primary lung cells, which is not the case. The authors should better indicate throughout the manuscript that they used a human lung carcinoma cell line.
Author Response
Reviewer 1
In this manuscript the authors intent to investigate the antiviral activity of carrageenans extracted from Halymenia floresii and Solieria chordalis against SARS-CoV-2. With the arising of the SARS-CoV-2 pandemic growing evidence manifested in the field that carrageenans, particularly iota-carrageenan, exhibit antiviral activity, demonstrated by different laboratories both, in vitro and in vivo. Furthermore, the antiviral activity of iota-carrageenan has been demonstrated before for a variety of respiratory viruses, including SARS-CoV-2. Also for lambda-carrageenan, an antiviral activity against SARS-CoV-2 was reported although proven less effective than iota-carrageenan. While iota-carrageenan cannot enter cells because of its molecular size, other like lambda carrageenan do penetrate cells and exert numbers of biological activities, among them immune activation. Most importantly, a multicenter, randomized, double-blinded, placebo-controlled clinical study revealed that an iota-carrageenan containing nasal spray exhibits prophylactic efficacy in preventing SARS-CoV-2 infection in healthcare workers caring for patients with COVID-19 disease. Another clinical trial investigated a nasal spray containing both, Ivermectin and iota-carrageenan showed reduction in COVID-19 as well as of disease severity.
While this manuscript might add additional information to the growing knowledge about antiviral activities of polysaccharides, the almost complete disrespect towards state of the published work prohibits any consideration about publishing this work as it was submitted for peer review and needs major revisions. Finally, the manuscript should be shortened to a note, as the only new information considers the anti-SARS-CoV-2 activity from the red algae Halymenia floresii and Solieria chordalis, which was not shown before, while nothing is new about the activities of iota- and lambda-carrageenans reported in this submission. Moreover, the quality as well as the interpretation of the data requires significant improvement.
Major
- As depicted in figure 1 and 2 the authors conducted spectroscopic and 1H NMR analysis. However, there are some issues with these analyses. First, the extracted fractions from Halymenia floresii and Solieria chordalis represent not an homogenous composition but rather a mixture of anything, among evidently iota- and lambda-carrageenan. Second, FTIR and 1H NMR analysis are not suitable to absolutely define the molecular structure of polymers like carrageenans, particularly if heterogeneous fractions are investigated. In fact, the authors could just make valuable conclusions about the antiviral efficacy of carrageenans, when there is a clear molecular characterization of the used fractions. At the moment they cannot define the nature and the content of the polymers in the extracts investigated.
We agree that FTIR and 1H NMR analysis are not suitable to define the absolute molecular structure. However, these studies have made it possible to distinguish between different types of polysaccharides in seaweeds, and have been widely used and validated based on specific signals allow distinguishing the main types of carrageenan (Chopin et al., 1999; Pereira et al., 2009; Gómez-Ordóñez et al., 2012). Further, the composition of the samples is not homogeneous due to the nature of the seaweeds, such as seasonality, environmental conditions, life cycle and/or the different batches of algal material (harvested, cultivated and/or strandings). The genus Solieria contains iota-carrageenans as its main cell wall polysaccharide (Deslandes et al., 1985. Saito and de Oliveira 1980, Murano et al., 1997), and the genus Halymenia contains mainly lambda-family carrageenans (Fenoradosoa et al., 2009; Freile-Pelegrin et al., 2011; Pliego-Cortés et al., 2022). Therefore, we have replaced the statements “lambda- and iota-carrageenan” by "polysaccharide-rich fraction in lambda- or iota-carrageenan". We have also added a paragraph in discussion (lines 414 to 432) based on the solubility of H. floresii and S. chordalis polysaccharides in potassium chloride at room temperature, according to our previous study in Pliego-Cortés et al. (2022). The H. floresii extracts were soluble in 0.3 M KCl solution and did not form any gel, after paper filtered the polysaccharides were recovered from the solution using isopropanol. While, S. chordalis extracts were insoluble and formed a soft gel similar to the commercial iota-carrageenan (Satiagel DF 52, SKW Biosystems, France). These results supports the predominantly occurrence of carrageenans belonging to lambda-family in H. floresii, since iota-carrageenans are known to be insoluble in 0.3 M KCl. Non homogeneous samples of lambda-carrageenans were also reported for their antiviral activity against SARS-CoV-2, in the study of Morokutti-Kurza et al., (2021), the authors showed by 1H NMR that lambda-carrageenan sample contained mainly kappa- and iota-carrageenan (41.6 and 27.3 % respectively), and in the study of Froba et al., (2021) the authors highlight that carrageenan homopolymers do usually not occur isolated in nature, and many of the available preparations contain relevant amounts of the other polymers. In both studies the authors used lambda-carragenans from DuPont (FMC Biopolymers, Philadelphia, PA, USA). Similarly, the study of Jang et al., (2021) used lambda-carragenans purchased from DuPont Nutrition & Biosciences (Wilmington, DE, USA) with an average molecular weight about 1025 kDa.
Chopin, T., Kerin, B. F. and Mazerolle, R. (1999). Phycocolloid chemistry as taxonomic indicator of phylogeny in the Gigartinales, Rhodophyceae: A review and current developments using Fourier transform infrared diffuse reflectance spectroscopy. Phycological Research, 47, 167-188.
Pereira L., Amado A., Ribeiro-Claro P., van de Velde F. VIBRATIONAL SPECTROSCOPY (FTIR-ATR AND FT-RAMAN): A Rapid and Useful Tool for Phycocolloid Analysis. 2009. BIODEVICES 2009 - International Conference on Biomedical Electronics and Devices.
Gómez-Ordóñez, E.; Jiménez-Escrig, A.; Rupérez, P. Molecular Weight Distribution of Polysaccharides from Edible Seaweeds by High-Performance Size-Exclusion Chromatography (HPSEC). Talanta 2012, 93, 153–159.
- Throughout the whole manuscript, the authors mainly use the term carrageenan, when they describe their own and also published work about the antiviral activity of carrageenans. E.g. they stated on page 2, line 70-75: “They also display antiviral properties, which have been demonstrated against several enveloped viruses including a wide range of respiratory viruses such as paramyxoviruses, respiratory syncytial virus (RSV), influenza viruses (FluV) and seasonal coronaviruses [13-21]. Recently, Pliego-Cortés et al. (2022) added in vitro evidence of the antiviral activity of carrageenans extracted from stranding biomass of Halymenia floresii and Solieria chordalis against Herpes simplex virus 1 (HSV-1) [22].” The authors should clearly indicate throughout the manuscript which type of carrageenan was used in these published studies and also what kind of carrageenans are used in the their own work.
As requested, we indicated throughout the manuscript which type of carrageenans was used in the published studies as well as in our work.
In this regard, it is well known from previous published work that there are significant differences in the antiviral efficacy of the different carrageenan-types. In general, iota-carrageenan shows a better antiviral effect than lambda- or kappa-carrageenan. In this context, it is also surprising that the extract from S.cordalis, which, according to the authors, contains iota-carrageenan does not have any antiviral effect, whereas H. floresii, which contains lambda-carrageenan inhibits virus replication. This is in contrast to the previously published work, especially regarding SARS-CoV-2.
No it is the opposite, it has been shown more often that lambda carrageenans are more effective than iota carrageenans for the main reason that they are richer in sulphate groups. However, it is necessary to be prudent. There are variations in chemical composition that can be observed over the year and according to geographical location. Moreover, sulphate groups are not the only elements linked to activity, the degree of sulfatation, distribution of sulfate groups, molecular weight, constituent sugars, conformation and dynamic stereochemistry, effect of counter cations can play a role. They do not always have the same mechanism of action, and the activity can be different depending on the moment of addition, well-illustrated in Carlucci's publication, in which the lambda-carrageenan fraction showed the most effective antiviral activity against HSV-1, however it is important to highlight that authors mentioned that the structure of this fraction seems to be responsible for its differential behavior compared with kappa/iota or and mu/nu carrageenans, even the cyclization of lambada-carrageenan affected their virucidal properties, since the lambda carrageenan composition and structure was altered. Regarding anti-SARS-CoV-2 activity, the study of Jang et al. (2011) using a lambda-carrageenan, with an average molecular weight 1025 kDa, the antiviral activity showed an EC50 of 0.9 ± 1.1 μg/ml, while the Ribavirin had an EC50 of 23.5 ± 1.2 μM, the authors successfully observed that λ-CGN inhibits not only SARS-CoV-2but also influenza viruses by targeting their entry process. Even, its virucidal properties led to a 60% survival rate in virus-challenged mice after an exposure of infectious virus to the antiviral agent.
Bourgougnon, N.; Chermann, J.C.; Lahaye, M.; Kornprobst, J.M. Anti-HIV activity and mode of action, in vitro, of the sulfated polysaccharide from Schizymenia dubyi (Rhodophyta). Cell. Pharmacol. 1996, 3, 104–108.
Carlucci, M.J.; Ciancia, M.; Matulewicz, M.C.; Cerezo, A.S.; Damonte, E.B. Antiherpetic activity and mode of action of natural carrageenans of diverse structural types. Antiviral Res. 1999, 43, 93–102.
Álvarez-Viñas, M.; Souto, S.; Flórez-Fernández, N.; Torres, M.D.; Bandín, I.; Domínguez, H. Antiviral Activity of Carrageenans and Processing Implications. Mar. Drugs 2021, 19, 437. https://doi.org/ 10.3390/md19080437
Morokutti-Kurz, M.; König-Schuster, M.; Koller, C.; Graf, C.; Graf, P.; Kirchoff, N.; Reutterer, B.; Seifert, J.M.; Unger, H.; Grassauer, A.; et al. The intranasal application of Zanamivir and carrageenan is synergistically active against influenza A virus in the murine model. PLoS ONE 2015, 10, e0128794.
This issue is also not discussed by the authors correctly. They stated in the discussion, page 9, line 303-304: “H. floresii contain mainly lambda-carrageenans and S. chordalis iota-carrageenans, the former being more active.” This statement is in contrast to their own work, were they show that iota-carrageenan is less active than lambda-carrageenan (Fig. 4).
In the sentence lines 303-304, “the former being more active” refers to lambda-carrageenans and therefore does not contradict our work, which indicated that lambda-carrageenans from H. floresii were more active than iota-carrageenans extracted from S. chordalis.
To avoid further confusion, we modified the sentences and therefore the discussion on this part. Also we included the studies related to antiviral activity against SARS-CoV-2, as in discussion starting in line 549: “Finally, the absence of anti-SARS-CoV-2 activity of fractions extracted from S. chordalis, rich in iota-carrageenans, assessed in the present work, contradicts previous data having demonstrated their efficacy in vitro as well as in nasal spray formulations [44, 45, 46]. It could be explained by the presence of a high protein content in S. chordalis fractions linked to the Enzyme-Assisted Extraction method with protamex. This mixture of proteases is known to increase protein recovery, when extracting polysaccharide fractions from S. chordalis, with amounts ranging from 11.6 to 15.2 % dry weight [30,31]. Proteins can be then involved in electrostatically-driven interactions with negatively-charged polysaccharides resulting in a coacervation and the formation of macromolecular complexes, significantly altering the structure of junction zones [47].”
Makshakova O.; Zuev Y. Interaction-Induced Structural Transformations in Polysaccharide and Protein-Polysaccharide Gels as Functional Basis for Novel Soft-Matter: A Case of Carrageenans. Gels 2022, 8, 287. https://doi.org/10.3390/gels8050287
- In figure 3 the authors analyzed the influence of the used fractions from red algae on the cell viability. The results of these experiments are somehow confusing. First, it seems that the treatment of the Calu-3 cells with the different fractions has a clear influence on the cell viability, inasmuch as the viability is increasing up to 200%. Have the authors any explanation for this observation? Moreover, the presented results just represents the results of two independent experiments. It is common practice in science to perform at least three independent experiments, also in order to be able to calculate significance. Second, the author should use a positive control for the induction of apoptosis, e.g. staurosporine.
This experiment demonstrated the absence of cytotoxicity of the fractions obtained from H. floresii and S. chordalis at the concentrations tested. We did not further investigate the effects of carrageenans on Calu-3 cells. Two independent experiments, each in duplicate, have been performed which corresponds to four evaluations of the cytotoxicity of each fraction. Cell viability was assessed using the cell proliferation kit II (XTT; Roche Diagnostics, Meylan, France) in accordance with the manufacturer’s instructions and compared with non-treated cells. We have validated this kit in several publications, which dispenses us with the use of a positive control.
- In figure 4 the authors used Heparin as a control, which is inappropriate. In their work, they mainly intend to show, that lambda- or iota-carrageenan from Halymenia floresii and Solieria chordalis inhibits the replication of SARS-CoV-2. It is published in several papers from several groups, that both, iota- and lambda-carrageenan exhibits antiviral activity against SARS-CoV-2. Thus, the well-characterized and – in addition - worldwide available carrageenans iota and lambda should be used as appropriate controls to demonstrate the efficacy in the inhibition of SARS-CoV-2 replication.
Heparin is also a sulfated polysaccaride (similar to those studied here) that acts as a bait to interfere with the binding of the spike protein (SARS-CoV-2) to the ACE2 receptor in the same way as carrageenans. It demonstrated effective inhibition of SARS-CoV-2 entry and therefore can be considered as appropriate control. Moreover, we thought it would be interesting to compare the effectiveness of our carrageenans with that of a commercially available reference molecule. Finally, the commercial carrageenan are not necessarily a better control since they are a mixture of different carrageenan, such as in the study of Morokutti-Kurza et al., (2021), the authors showed by 1H NMR that lambda-carrageenan sample contained mainly kappa- and iota-carrageenan (41.6 and 27.3 % respectively).
- In figure 4 the authors measured viral RNA in the cell supernatant as well as in the cell monolayer. Why did they measured the content of the viral RNA intracellularly? Moreover, the labeling of the figures is confusing. In Figure 4 A, the authors used the term “viral copies/ng RNA” in figure 4 B, they used “viral copies/mL” and in figure 4 C: “Viral titer”. What do the authors mean with the term “viral titer”? Overall, the authors should choose a comprehensible and unique labeling.
We measured the viral RNA intracellularly to assess the effect of carrageenans on virus replication while viral RNA quantification in the cell culture supernatant shows both intracellular replication and release of viral particles. From the cell monolayer, the viral load was normalized per nanogram of total RNA extracted while in the supernatant normalization was done per mL of culture medium. Finally, in addition to the viral load measured by RT-qPCR, we also titrated the virus in order to demonstrate an impact of the molecules tested on the production of infectious viral particles.
- In figure 5 the authors conducted time of addition experiments. First of all, it would be advantageous for the fast understanding of this series of experiments to integrate a schematic depiction of the different experimental setups in the figure.
Coming to the results of this figure, there were no big differences between the different treatment options. Clear conclusions from the authors according to these results are missing, because they might be impossible. The authors do not write anything in the abstract nor in the results section about their conclusions regarding the results of these experiments. Since the pre-treatment shows a very similar result on the inhibition of virus replication as the treatment after infection with SARS-CoV-2, it might be concluded that carrageenans could be used in both cases: before an infection, i. e. prophylactically, and after an infection, i. e. therapeutically. However, this has not been shown before, even not in the clinical trials.
Even worse, in the paragraph in the results section concerning these experiments, the authors wrote on page 6, line 205-207: “Finally, the activities of the two fractions 1.1 and 1.3 were compared with each other. Overall, fraction 1.1 exhibited significantly higher antiviral activity against SARS-CoV-2 than fraction 1.3.”. However, the authors did not directly compare the different fractions, at least not within the depicted figure 5, thus they could not conclude from this figure, that fraction 1.1 indeed exhibits significant higher antiviral activity compared with fraction 1.3.
A schematic depiction of the different experimental setups was added in the figure 5 (Figure 5A). The fact that our results suggest a possible use of carrageenans both prophylactically and curatively was added to the discussion. Finally, the sentence lines 205-207, page has been modified to indicate similar activity of the two fractions.
- In figure 6 the authors intend to analyze the immunomodulatory properties of the H. floresii fractions. As stated by the authors, these fractions contain lambda-carrageenan. It was published previously in countless publications that lambda carrageenan has immunomodulatory properties, e.g. it could induce inflammation, which is in a clear contrast to the results of these manuscript. Thus, the authors should conduct more thorough tests or entirely remove this figures and the conclusions drawn from these experiments.
In order to respond to the comments made by the two reviewers, figure 6 as well as the conclusions drawn from this figure have been removed.
- In figure 7 the authors wanted to investigate if the different fractions of H.floresii have any influence on the SARS-CoV-2 attachment to the cells. Thereby, they concluded from their experiments on page 8, line 243-244: ”The viral adsorption assay showed a reduction of SARS-CoV-2 replication under the three protocols and for the two fractions tested.” Looking at the figure, it seem that in the most cases there is no significant reduction of virus replication. Thus, the authors could only make pure speculations, that the binding of the virus to the cells is affected. This conclusion is not supported by the results of the conducted experiments. The authors should remove these speculations or perform experiments, which clearly support their conclusions.
In figure 7 (became figure 6 after removal of experiments about immunomodulation), viral replication is decreased in all conditions tested. This inhibition of virus replication was even statistically significant when viral RNA was quantified in the cell culture supernatant. However, an action of H. floresii lambda-carrageenans on the SARS-CoV-2 attachment to the cells is indeed an hypothesis deduced from the results of this experiment and presented as such in the discussion. It nevertheless confirms the data previously published in the literature on the mode of antiviral action of carrageenans.
Minor
- Within the title the authors state: “Anti-SARS-CoV-2 activity of polysaccharides extracted from Halymenia floresii and Solieria chordalis (Rhodophyta).” This title is misleading as there are different variants of concern of SARS-CoV-2 (Alpha, Beta, Gamma, Delta and Omicron). However, the authors only investigated the influence of different polysaccharides on the replication of SARS-CoV-2 Wuhan Type, this should be clearly indicated in the title.
The type of SARS-CoV-2 used is clearly indicated in the introduction and in the materials and methods.
- In the introduction the authors should be clearer in the scientific classification of SARS-CoV-2 as well as the other endemic coronaviruses. Some belongs to the beta-coronaviruses and some of them to the alpha-coronaviruses.
Classification of coronaviruses infecting humans has been added to the introduction.
- In the introduction page 2, line 56-59 the authors stated: “Nevertheless, the rapid genetic evolution of SARS-CoV-2, through mutations and recombinations, is likely to cause the emergence of resistance to current antivirals. The search for new antiviral molecules to prevent and treat COVID-19 is therefore still relevant.” This is a wrong argument, as, except for monoclonal antibodies, no resistance mutations have been reported so far for any antiviral against SARS-CoV-2.
We are not saying that resistance to anti-COVID-19 treatment exists, only that it is likely to emerge at any time. Several publications have nevertheless already described in vivo as well as in vitro mutations likely to confer resistance to nirmatrelvir (Global prevalence of SARS-CoV-2 3CL protease mutations associated with nirmatrelvir or ensitrelvir resistance. Ip JD, Wing-Ho Chu A, Chan WM, Cheuk-Ying Leung R, Umer Abdullah SM, Sun Y, Kai-Wang To K. EBioMedicine. 2023 Apr 13;91:104559. doi: 10.1016/j.ebiom.2023.104559). Being able to rely on several antiviral molecules with different mechanisms of action is therefore a necessity.
- On page 2, line 92-94 the authors stated: “Evaluation of carrageenan treatment on viral replication was performed during kinetics of infection of human airway epithelial Calu-3 cells.” This sounds like they used primary lung cells, which is not the case. The authors should better indicate throughout the manuscript that they used a human lung carcinoma cell line.
Calu-3 cells are described as Human Airway Epithelial Cells on the ATCC website. Nevertheless, the mention of their isolation from lung cancer has been added in the introduction and the materials and methods.
Reviewer 2 Report
The field of anti-viral polysaccharides is constantly growing and the manuscript provides valuable new insight into the effect of extract from Halymenia floresii and Solieria chordalis against SARS-CoV-2. For publication substantial adaptations of the manuscript are required especially with respect to the context of the research and the conclusions drawn from the experiments.
1. Introduction:
While it is correctly mentioned and cited that several types of carrageenans are active against SARS-CoV-2 it is of particular importance that at least the carrageenan from Halymenia floresii has been characterized in the past (Freile-Pelegrin et al., 2011). As major part of the experiments refer to the enzymatic extraction method the authors may want to give some evidence why they did not use standard methods described for carrageenan extraction for quite some time
2. Results:
Figure 2 shows the 1HNMR spectra of two fractions of H. floresii. No further NMR data are provided and any conclusions for other fractions are not justified and literature should only be discussed in the results section when needed.
Figure 3: It seems that the untreated controls of the Calu-3 cells underwent a different treatment than the tested fractions at different concentrations. Even at a concentration of 5µg/ml the viability varies between 110-155% with huge standard deviations that may arise from the method to calculate the SEM of two independent experiments from duplicates. This method of calculation is not acceptable for publication and the Figure needs revision, with the untreated control cells shown in the figure as well.
Figure 4 and 5 As the authors have mentioned correctly in their introduction that carrageenans are already known for their activity against SARS-CoV-2 they failed to include a antiviral positive control in their experiment. This could be either iota or lambda- carrageenan from a commercial source. This would help put the findings into context of existing, published data and validate their own work.
Figure 6
The name of the figure is misleading, and the conclusion drawn from it later in the discussion are false. The experiment only shows the m-RNA levels of certain biomarkes after 4 hours of incubation on the cells and not more. There are more than 5000 hits with the search term carrageenan and inflammation in pubmed as especially lambda-carrageenan is commonly used in the carrageenan induced paw edema model. The authors are encouraged to either deepen their understanding about the inflammatory properties of their extracts or simply remove the experiment and the conclusions from the manuscript.
3. Discussion:
On page 9 starting from line 290 the authors begin to discuss and speculate around the antiviral activity and the sulfate content. For that paper the authors refer to a publication from 2009 long before the COVID-19 pandemic and fail to take recent SARS-CoV-2 related literature into account. In the sentence at line 303 the authors state that H. floresii contains mainly lambda carrageenan and S. Chordalis iota carrageenan, with the latter being more active. This statement is in contrast to data presented in the manuscript. First there is no data showing that iota carrageenan is present at least no 1H NMR data and second the authors show the contrary in Fig.4. This is entirely confusing and should be clarified.
However, starting from line 404 unjustified speculation continues with absolutely no context to the manuscript that deals with the antiviral activity against SARS-CoV-2. Instead of speculating around with publication that never dealt with a pandemic Coronavirus the authors should focus on putting their own data into context with previously published data.
Later the authors try to digest from literature but not from own data, how the chain-length or the molecular weight of a polymer influence the antiviral activity, again based on decade old literature and out of context with SARS-CoV-2. Instead of speculation the authors are encouraged to conduct experiments with degraded polymers either from their own extracts or from commercially available carrageenan with SARS-CoV-2. For viruses other than SARS-CoV-2 it has been shown and published that degradation of polymers leads to the loss of antiviral activity leading to complete inactivity of monomeric or dimeric sugars. Hence, the conclusion that the poor tissue-penetrating ability can be an obstacle needs substantial revision as well or even better support from own data.
Besides that several recent publications relating to SARS-CoV-2 and carrageenan are not mentioned and discussed including but not limited to a RCT clinical trial from Argentina that showed a prophylactic effect of iota-carrageenan (Figueroa et al., Int J Gen Med. 2021;14:6277-6286).
Hence, the discussion requires substantial revison.
4. Enzymatic degradation - supplementary material and conclusion drawn from line 361
The authors fail to provide data supporting their statement that enzymatic degradation is of any benefit at all. Given the results that S. chordalis extracts showed lower activity than H. floresii one could speculate also based on the MW data provided in Table 1 that H. floresii is just more resistant against enzymatic degradation leading to a higher molecular weight and therefore a better efficacy was found against SARS-CoV-2. However, more data would be needed to substantiate the speculation.
If the authors whish demonstrating any benefit of their extraction method, they should compare it with standard extraction methods known for carrageenans and provide material at a quality level as outlined for example in the US pharmacopeia for carrageenans.
Table S1 shows, that the extract fractions 1.1 to 1.6 contain between 1,1% and 7,5% protein The authors are encouraged to discuss the unusual high protein content, why protein is present at all and what consequences this may have. At least for application to humans this would be an issue.
Besides some typos in the discussion the English is clear and understandable.
Any issues can be easily sorted out during proofreading.
Author Response
Reviewer 2
The field of anti-viral polysaccharides is constantly growing and the manuscript provides valuable new insight into the effect of extract from Halymenia floresii and Solieria chordalis against SARS-CoV-2. For publication substantial adaptations of the manuscript are required especially with respect to the context of the research and the conclusions drawn from the experiments.
- Introduction:
While it is correctly mentioned and cited that several types of carrageenans are active against SARS-CoV-2 it is of particular importance that at least the carrageenan from Halymenia floresii has been characterized in the past (Freile-Pelegrin et al., 2011). As major part of the experiments refer to the enzymatic extraction method the authors may want to give some evidence why they did not use standard methods described for carrageenan extraction for quite some time.
As the reviewer requested, we added in the introduction the corresponding reference for the preliminary characterization of H. floresii carrageenan reported by Freile-Pelegrin et al., 2011. “The native sulfated polysaccharides of H. floresii were previously studied by Freile-Pelegrin and colleagues. The authors reported that these polysaccharides seemed to be chemically and rheologically similar to the lambda-carrageenan family [25].” We, also added the information for Solieria: “The genus Solieria contains iota-carrageenan as its main cell wall polysaccharide (Deslandes et al., 1985. Saito and de Oliveira 1980, Murano et al., 1997).”
Regarding to the enzymatic extraction, we have focused our work on developing new, more environmentally friendly extraction methods. Therefore, we included the following information in the introduction : “In this sense, the extraction efficacy must impact the potential utilization of the seaweed biomass. For instant, the enzyme-assisted extraction (EAE) is a green environmentally friendly extraction method known for its high efficiency and allowing to reduce, or avoid, the use of solvent or alkali, which could induce the chemical or structural modification (i.e. the cyclized derivatives in lambda-type carrageenans). Our research group showed the efficacy of the EAE procedure on native polysaccharide extraction yields from Chondrus crispus, Solieria filiformis and chordalis, and H. floresii using commercial enzymes, generally proteases or glucanases [22, 30-34].”
- Results:
Figure 2 shows the 1HNMR spectra of two fractions of H. floresii. No further NMR data are provided and any conclusions for other fractions are not justified and literature should only be discussed in the results section when needed.
Since the polysaccharide fractions rich in lambda carrageenan from H. floresii showed the highest antiviral activity, we only proceeded to obtain the spectra for this species, in order to display the chemical shifts related mainly to lambda-type carrageenans; the spectrum of the fractions obtained with hot water (HWE) was presented in our previous publication (Pliego- Cortes et al., 2022). Therefore, we have modified the text in the results section as follows: “Since the polysaccharide fractions rich in lambda-carrageenan obtained through the EAE from H. floresii showed higher antiviral activity, the structural composition of these polysaccharide fractions 1.1 and 1.3 were further explored by the 1H Nuclear Magnetic Resonance (NMR).”
While, for S. chordalis polysaccharides, the FT-IR allowed to successful identified that fractions 1.5 and 1.6 were iota-carrageenan-rich polysaccharide fractions. Moreover, we included a few lines of discussion about the solubility of H. floresii and S. chordalis polysaccharides in potassium chloride at room temperature: “Moreover, this study demonstrated the efficacy of the eco-friendly processes (Enzyme-Assisted Extraction) to increase the yield of extractions and prevent changes in the molecules, since polysaccharide extraction in red seaweeds is usually done with a strong hot-alkaline solution, which affects the structure and the size of the molecules, and also their biological properties [37]. Finally, H. floresii polysaccharides were soluble in 0.3 M KCl solution and recovered from the solution using isopropanol whereas S. chordalis extracts were insoluble and formed soft gels. These results supported the predominantly occurrence of carrageenans belonging to lambda-family in H. floresii polysaccharide fractions while iota-carrageenans were majority in S. chordalis [22].”
Figure 3: It seems that the untreated controls of the Calu-3 cells underwent a different treatment than the tested fractions at different concentrations. Even at a concentration of 5µg/ml the viability varies between 110-155% with huge standard deviations that may arise from the method to calculate the SEM of two independent experiments from duplicates. This method of calculation is not acceptable for publication and the Figure needs revision, with the untreated control cells shown in the figure as well.
The untreated controls of the Calu-3 cells do not appear in this figure because they are used as a reference to calculate the cell viability of the treated cells. The viability of the control cells therefore defines 100% viability. The viability of the treated cells is then statistically compared with that of the untreated control cells. A significant difference indicates cytotoxicity. In our study, no significant difference between treated and untreated cells was observed either up or down.
Figure 4 and 5 As the authors have mentioned correctly in their introduction that carrageenans are already known for their activity against SARS-CoV-2 they failed to include a antiviral positive control in their experiment. This could be either iota or lambda- carrageenan from a commercial source. This would help put the findings into context of existing, published data and validate their own work.
Heparin is also a sulfated polysaccaride (similar to those studied here) that acts as a bait to interfere with the binding of the spike protein (SARS-CoV-2) to the ACE2 receptor in the same way as carrageenans. It demonstrated effective inhibition of SARS-CoV-2 entry and therefore can be considered as appropriate control. Moreover, we thought it would be interesting to compare the effectiveness of our carrageenans with that of a commercially available reference molecule. Finally, the commercial carrageenan are not necessarily a better control since they are a mixture of different carrageenan, such as in the study of Morokutti-Kurza et al., (2021), the authors showed by 1H NMR that lambda-carrageenan sample contained mainly kappa- and iota-carrageenan (41.6 and 27.3 % respectively).
Figure 6
The name of the figure is misleading, and the conclusion drawn from it later in the discussion are false. The experiment only shows the m-RNA levels of certain biomarkes after 4 hours of incubation on the cells and not more. There are more than 5000 hits with the search term carrageenan and inflammation in pubmed as especially lambda-carrageenan is commonly used in the carrageenan induced paw edema model. The authors are encouraged to either deepen their understanding about the inflammatory properties of their extracts or simply remove the experiment and the conclusions from the manuscript.
In order to respond to the comments made by the two reviewers, figure 6 as well as the conclusions drawn from this figure have been removed.
- Discussion:
On page 9 starting from line 290 the authors begin to discuss and speculate around the antiviral activity and the sulfate content. For that paper the authors refer to a publication from 2009 long before the COVID-19 pandemic and fail to take recent SARS-CoV-2 related literature into account. In the sentence at line 303 the authors state that H. floresii contains mainly lambda carrageenan and S. Chordalis iota carrageenan, with the latter being more active. This statement is in contrast to data presented in the manuscript. First there is no data showing that iota carrageenan is present at least no 1H NMR data and second the authors show the contrary in Fig.4. This is entirely confusing and should be clarified.
In addition to the study by Ghosh et al, published in 2009, we have cited the study of Yim et al., (2021). These authors showed that crude polysaccharide from the red alga Porphyra tenera with undetectable amount of sulfate content had no antiviral activity against SARS-CoV-2 pseudovirus (COV-PS02, Creative Diagnostics); while Kang et al., (2022) reported that the polysaccharides with the highest content of sulfate ions (extracted from the brown alga Hizikia fusiforme) presented the highest anti SARS-CoV-2 activity; which confirms the positive correlation between the antiviral activity and the sulfate content.
In the sentence lines 303-304, “the former being more active” refers to lambda-carrageenans and therefore does not contradict our work, which indicated that lambda-carrageenans from H. floresii were more active than iota-carrageenans extracted from S. chordalis. However, to avoid further confusion, we modified the sentences and therefore the discussion on this part.
However, starting from line 404 unjustified speculation continues with absolutely no context to the manuscript that deals with the antiviral activity against SARS-CoV-2. Instead of speculating around with publication that never dealt with a pandemic Coronavirus the authors should focus on putting their own data into context with previously published data.
Overall, the literature describing the antiviral activities of carrageenans, even against other viral species than SARS-CoV-2, is relevant because the antiviral mechanisms involved are probably identical. Additionally, the only non-SARS-CoV-2 virus discussed here is HSV-1 against which the antiviral effects of H. floresii lambda-carrageenans and S. chordalis iota-carrageenans were recently tested.
Later the authors try to digest from literature but not from own data, how the chain-length or the molecular weight of a polymer influence the antiviral activity, again based on decade old literature and out of context with SARS-CoV-2. Instead of speculation the authors are encouraged to conduct experiments with degraded polymers either from their own extracts or from commercially available carrageenan with SARS-CoV-2. For viruses other than SARS-CoV-2 it has been shown and published that degradation of polymers leads to the loss of antiviral activity leading to complete inactivity of monomeric or dimeric sugars. Hence, the conclusion that the poor tissue-penetrating ability can be an obstacle needs substantial revision as well or even better support from own data.
Besides that several recent publications relating to SARS-CoV-2 and carrageenan are not mentioned and discussed including but not limited to a RCT clinical trial from Argentina that showed a prophylactic effect of iota-carrageenan (Figueroa et al., Int J Gen Med. 2021;14:6277-6286).
Hence, the discussion requires substantial revison.
The second part of the conclusion mentioning the interest of working with smaller molecules in order to facilitate tissue penetration has been removed from the article. Conversely, the study by Figueroa et al. has been added to the discussion.
- Enzymatic degradation - supplementary material and conclusion drawn from line 361
The authors fail to provide data supporting their statement that enzymatic degradation is of any benefit at all. Given the results that S. chordalis extracts showed lower activity than H. floresii one could speculate also based on the MW data provided in Table 1 that H. floresii is just more resistant against enzymatic degradation leading to a higher molecular weight and therefore a better efficacy was found against SARS-CoV-2. However, more data would be needed to substantiate the speculation.
If the authors whish demonstrating any benefit of their extraction method, they should compare it with standard extraction methods known for carrageenans and provide material at a quality level as outlined for example in the US pharmacopeia for carrageenans.
Table S1 shows, that the extract fractions 1.1 to 1.6 contain between 1,1% and 7,5% protein The authors are encouraged to discuss the unusual high protein content, why protein is present at all and what consequences this may have. At least for application to humans this would be an issue.
As mentioned in the answer done for the introduction, the enzymatic extraction (EAE) was carried out to perform an environmentally friendly extraction to obtain native carrageenan, avoiding the use of alkali, which could induce structural and biochemical modification, or solvent residues. The comparison with one of the traditional extraction methods was the hot water extraction, this allowed us to corroborate the efficiency of the EAE mainly based on terms of extraction yields. In our previous study (Pliego-Cortes et al., 2022), we showed that the EAE allowed a significantly higher extraction yield of the polysaccharides of H. floresii and S. chordalis (c.a. 20% of yield in dry weight, after EtOH and dialysis isolation) in addition, as we mentioned previously, other studies in our research group have shown the EAE advantage in extraction yields (Hardouin et al., 2014; Kulshreshtha et al., 2015; Burlot et al., 2016; Penuela et al., 2020; Spain et al., 2022). The protamexR enzyme (Novozymes) used is a commercially available mixture of microbial proteases which have been used in seaweeds to release bioactive compounds, the enzyme do not have specific activity on carrageenans, the enzyme acts on the seaweed cell wall, breaking the link between proteins and carbohydrates. So, the enzyme disrupts and fractionates the cell wall allowing the extraction of their components. In general, carrageenans has been reported with a wide MW distribution with an average MW up to 800 kDa as reported by McKim et al. (2019). In our results, the MW of the different fractions are in this range, and the differences between H. floresii and S. chordalis sample could be mainly related based on McKim et al. (2019) explanation. The MW profile represents a mean of capturing an average molecular weight of the carrageenans at the time of seaweed harvesting, and seaweeds may generate molecules of different sizes depending on their growth needs.
Although we did not evaluate the quality of the carrageenans based on the US pharmacopeia, we applied one of the protocols reported to obtain them. According to the US pharmacopeia, carrageenans are extracted with water or aqueous alkali, and recovered by alcohol precipitation. This alcohol is restricted to methanol, ethanol or isoproil alcohol. Regarding to the enzyme used, Protamex® (Novozymes) is a mixture of microbial proteases, preservative-free and suitable for organic products, for use in infant formula, or for kosher and halal products. This allowed us to ensure that the enzyme does not represent an problem in the quality of the extracted polysaccharides. In addition, the enzyme was eliminated during the precipitation of the polysaccharide with alcohol and its subsequent dialysis process.
Since protamex enzyme is a mixture of proteases, it is possible that protamex allowed a better protein extraction from S. chordalis. Hardouin et al. reported that proteases allowed a recovering of protein between 11.6 and 15.2 % of S. chordalis dry weight. While Burlot et al., by employing a response surface methodology, found the best conditions for EAE application to obtain water-soluble compounds in Solieria chordalis, the authors reported that extracts, after the action of enzymes and after the optimization of extraction conditions, were especially rich in proteins (14.7 %). Furthermore, the antiviral activity of the extracts was kept on HSV-1.
We have modified the discussion in this sense.
Hardouin, K.; Burlot, A.-S.; Umami, A.; Tanniou, A.; Stiger-Pouvreau, V.; Widowati, I.; Bedoux, G.; Bourgougnon, N. Biochemical and Antiviral Activities of Enzymatic Hydrolysates from Different Invasive French Seaweeds. J. Appl. Phycol. 2014, 26, 1029–1042.
Kulshreshtha, G.; Burlot, A.-S.; Marty, C.; Critchley, A.; Hafting, J.; Bedoux, G.; Bourgougnon, N.; Prithiviraj, B. Enzyme-Assisted Extraction of Bioactive Material from Chondrus Crispus and Codium Fragile and Its Effect on Herpes Simplex Virus (HSV-1). Mar. Drugs 2015, 13, 558–580.
Burlot; Anne-Sophie.; Gilles, B. Response Surface Methodology for Enzyme-Assisted Extraction of Water- Soluble Antiviral Compounds from the Proliferative Macroalga Solieria Chordalis. Enz. Eng. 2016, 5, 1000148.
Peñuela, A.; Nathalie, B.; Gilles, B.; Daniel, R.; Tomás, M.-S.; Yolanda, F.-P. Anti-Herpes Simplex Virus (HSV-1) Activity and Antioxidant Capacity of Carrageenan-Rich Enzymatic Extracts from Solieria Filiformis (Gigartinales, Rhodophyta). Int. J. Biol. Macromol. 2021, 168, 322–330.
Spain, O., Hardouin, K., Bourgougnon, N. et al. Enzyme-assisted extraction of red seaweed Solieria chordalis (C.Agardh) J. Agardh 1842—the starting point for the production of biostimulants of plant growth and biosorbents of metal ions. Biomass Conv. Bioref. (2022). https://doi.org/10.1007/s13399-022-02456-7
J.M. McKim, J.A. Willoughby Sr., W.R. Blakemore, M.L. Weiner, Clarifying the confusion between poligeenan, degraded carrageenan, and carrageenan: a review of the chemistry, nomenclature, and in vivo toxicology by the oral route, Crit. Rev. Food Sci. Nutr. 59 (2019) 3054–3073, https://doi.org/10.1080/10408398.2018.1481822.
Round 2
Reviewer 1 Report
I accept it with the statement, that the issue about antiviral activity of iota- versus lambda-carrageenan against SARS-CoV-2 still has not been carefully elaborated, however, in the interest of getting this work accessible to the community I accept publication .
Author Response
We thank the reviewer for her/his comments which allowed us to improve the quality of our work.
Reviewer 2 Report
While some of the confusions have been removed the authors still try to draw conclusions that are not supported by data. In the entire manusscript there is not a single datapoint proving that iota-carrageenen was tested at all or was present in the S.chordalis extract. However, the authors try to speculate around the activity of carrageenans without any controls with pure substances. I suggest to remove the speculations and reduce the manuscript to to the data that are shown.
To make it short. The authors were able to show that an extract from H. floresii bears anti-SARS-CoV-2 effects. They show that lambda-Carrageenan is present in this extract and were able to confirm previous studies on Carrageenan with a potential new method of extraction. More data are not provided.
The authors failed to adress the SEM issue with Figure 3. Calculating a SEM from two variables is simply not statistically meaningful.
Author Response
While some of the confusions have been removed the authors still try to draw conclusions that are not supported by data. In the entire manusscript there is not a single datapoint proving that iota-carrageenen was tested at all or was present in the S. chordalis extract*. However, the authors try to speculate around the activity of carrageenans without any controls with pure substances. I suggest to remove the speculations and reduce the manuscript to the data that are shown. To make it short. The authors were able to show that an extract from H. floresii bears anti-SARS-CoV-2 effects. They show that lambda-Carrageenan is present in this extract and were able to confirm previous studies on Carrageenan with a potential new method of extraction. More data are not provided.
We have performed and added the FT-IR spectra for commercial samples of lambda- and iota-carrageenan in figure 1 as a point of reference. When comparing our samples with the commercial samples, we have obtained that the typical bands for Solieria carrageenans correspond to the commercial iota-carrageenans. Joint to this, the behavior of the Solieria' extracts in the potassium solution (producing soft gel) and the high occurrence of the galactose monosaccharide and low glucose content, support that the predominant carrageenan is of iota type. Bondu et al. (2010), confirmed by FTIR, 13C NMR and GC-MS that the main type of carrageenan in S. chordalis from Brittany, extracted with hot water, was the iota type, with minor content of pyruvated alpha- and nu-carrageenan. While Fournet et al. (1997) determined in situ that the main carrageenan presented in the cellular wall of S. chordalis was iota-type. Finally, Prado-Fernandez et al (2003) published a well-established methodology with FT-IR, evaluated 160 mixtures and managed to differentiate between the different types of iota, kappa and lambda carrageenans, showing the validity of FT-IR analysis.
Fournet, E. Ar Gall, E. Deslandes, J.-P. Huvenne, B. Sombret, J.Y. Floc'h. In situ Measurements of Cell Wall Components in the Red Alga Solieria chordalis (Solieriaceae, Rhodophyta) by FTIR Microspectrometry. Bot. March 40 (1997) 45.
Prado-Fernández, J.A. Rodriguez-Vazquez, E. Tojob, J.M. Andrade. Quantitation of k-, i- and l-carrageenans by mid-infrared spectroscopy and PLS regression. Analytica Chemical Act 480 (2003) 23–37e
The authors failed to adress the SEM issue with Figure 3. Calculating a SEM from two variables is simply not statistically meaningful.
It is true that this experiment was only performed twice because our goal was to revalidate in our hands the absence of cytotoxicity of the extracts of Halymenia floresii and Solieria chordalis carrageenans already demonstrated in the study by Pliego-Cortés et al. 2022 (doi.org/10.3390/md20020116). In this article, an absence of toxicity was demonstrated on Vero cells up to a concentration of 1000 μg/mL. By comparison, the concentration used in the present work of 10 µg/mL is very low. Finally, even though we only performed two replicates, each experiment included 2 cytotoxicity assessments on two different wells. Our results are therefore based on 4 variables.